# TimeSAE: Causal Sparse Decoding for Faithful Explanations of Black-Box Time Series Models

Khalid Oublal*[1]   Quentin Bouniot[1 2 3 4 5]   Qi Gan[1]   Stéphan Clémençon[1]   Zeynep Akata[2 3 4 5]

## Abstract

As black box models and pretrained models gain traction in time series applications, understanding and explaining their predictions becomes increasingly vital, especially in high-stakes domains where interpretability and trust are essential. However, most of the existing methods involve only in-distribution explanation, and do not generalize outside the training support, which requires the learning capability of generalization. In this work, we aim to provide a framework to explain black-box models for time series data through the dual lenses of Sparse Autoencoders (SAEs) and causality. We show that many current explanation methods are sensitive to distributional shifts, limiting their effectiveness in real-world scenarios. Building on the concept of Sparse Autoencoder, we introduce `TimeSAE`, a framework for black-box model explanation. We conduct extensive evaluations of `TimeSAE` on both synthetic and real-world time series datasets, comparing it to leading baselines. The results, supported by both quantitative metrics and qualitative insights, show that `TimeSAE` delivers more faithful and robust explanations. Our code and dataset are available in an easy-to-use library `TimeSAE`-Lib: https://oublalkhalid.github.io/TimeSAE/.

## 1. Introduction

The rise of black box models such as large foundation models has revolutionized various fields, including time series analysis, with applications in finance (Bento et al., 2021), healthcare (Kaushik et al., 2020), and environmental science (Adebayo et al., 2021). These networks often make

critical decisions, especially in sensitive domains where decisions are based on forecasting outcomes, such as managing grid stability in Energy (Eid et al., 2016), and Healthcare (Dairi et al., 2021), yet the underlying decision-making process is difficult to interpret due to the black-box nature of the models. This opacity has motivated the rise of explainable AI (XAI) techniques to provide human-understandable explanations for model decisions. While XAI has been predominantly applied in image classification, it is extending into other fields, such as audio and time series (Parekh et al., 2022; Queen et al., 2023).

Current methods in enhancing explainability for time series primarily identify key signal locations (sub-instance) affecting model predictions. For instance, Shi et al. (2023) uses LIME (Ribeiro et al., 2016) to explain water level prediction models. Additionally, perturbation methods like Dynamask (Crabbé & Van Der Schaar, 2021) and Extrmask (Enguehard, 2023) modify less critical features to evaluate their impact but often struggle with feature interdependencies and generalization. Despite their insights, these techniques face challenges with Out-of-Distribution (OOD) samples, affecting the *faithfulness* of explanations (Queen et al., 2023).

Explaining time series black-box models requires the ability to generalize beyond the training distribution, which is essential for the robust deployment of explanatory algorithms in real-world scenarios. In addressing the extrapolation of explanation, Queen et al. (2023) retrain a white-box model for consistency, though this depends on knowing the model's structure and may not ensure consistent explanations. Similarly, Liu et al. (2024b) uses a stochastic mask to tackle OOD issues; however, challenges remain as explanations are still treated as OOD and lack clarity, with explanation compositionality aspects unaddressed. Furthermore, *faithfulness* is also a desirable property of any explanation method, broadly defined as the ability of the method to provide accurate descriptions of the underlying reasoning (Gat et al., 2023; Jain & Wallace, 2019). Formally, it can be defined as:

**Definition 1** (**Faithful Explanations.**). *Let* $f : \mathcal{X} \to \mathcal{Y}$ *be a black-box model, consider an input* $\mathbf{x} \in \mathcal{X}$, *and let* $\mathcal{E} : \mathcal{X} \to \mathcal{C}$ *be an explainer that generates explanations in the concept space* $\mathcal{C}$. *Faithfulness is the ability of* $\mathcal{E}$ *to accurately reflect* $f$'s *reasoning, i.e the predictive capability*

---

[1]LTCI, Télécom Paris, Institut Polytechnique de Paris [2]Technical University of Munich [3]Helmholtz Munich [4]Munich Center for Machine Learning [5]Munich Data Science Institute. Correspondence to: Khalid Oublal <khalid.oublal@ip-paris.fr>, Quentin Bouniot <quentin.bouniot@telecom-paris.fr>.

*Proceedings of the 43rd International Conference on Machine Learning*, Seoul, South Korea. PMLR 306, 2026. Copyright 2026 by the author(s).

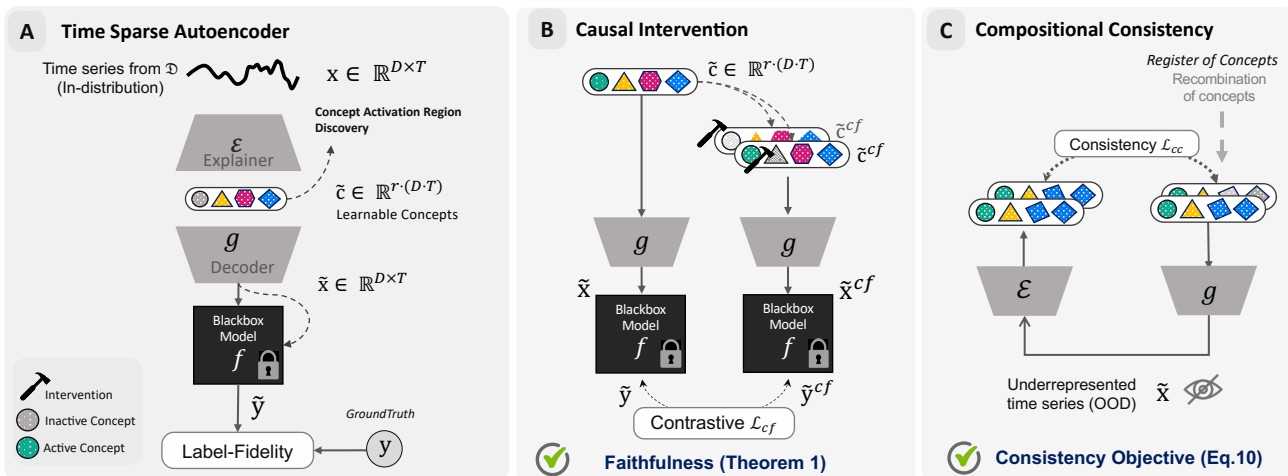

**Figure 1.** **Overview of Time Series Sparse Autoencoder ( `TimeSAE` ): (A)** The framework assumes access to a black-box model $f$ and aims to explain its predictions on data $x \in \mathcal{X}$ by learning an encoder $\mathcal{E}$ and a decoder $g$ that decompose the time series into interpretable components. **(B)** For faithfulness, the sparse autoencoder $(\mathcal{E}, g)$ incorporates properties to leverage counterfactual explanations. A contrastive learning loss incorporates a set of counterfactuals $\tilde{x}^{cf}$, obtained by intervening on concepts and which produce, via $f$, a contradictory label $\tilde{y}^{cf}$ relative to the original $\tilde{y}$. **(C)** To ensure compositional explanations, the method enforces the encoder $\mathcal{E}$ to generate consistent explanations by combining intermediate findings.

*of $f(\mathbf{x})$ from the explanations $\mathcal{E}(\mathbf{x})$.*

To address Definition 1, there is often confusion between two types of model explanations: **a)** *what knowledge does the model encode?* **b)** *Why does it make certain predictions?* Here, the *"what"* aspect does not directly explain the model's decision-making, as not all encoded features are necessarily used in predictions. Explanations based on the *"what"* are typically correlational, rather than causal. Understanding a model's reasoning requires attention to the *"why"*, making causality indispensable, which is defined as how changes in inputs impact the outputs. Indeed, counterfactuals (CFs), which explore *"what if"* scenarios by identifying minimal and feasible changes to inputs that alter predictions, are at the highest level of Pearl's causal hierarchy (Pearl, 2009), highlighting how changes lead to a different prediction. To truly understand a model's reasoning, it is essential to focus on the *"why"*, making causality a key component of *faithfulness*.

**Contributions** We propose `TimeSAE` , a Sparse Autoencoder (SAE) for learning explanation for time series black-box models. Specifically, we replace the need of the sub-instances via masking time steps and features by directly learning an end-to-end SAE using JumpReLU (Rajamanoharan et al., 2024b) to explain the prediction by concepts, using explaination-embedded instances that are close to the original distribution while maintaining label-prediction. Our method is model-agnostic and operates in a post-hoc manner, requiring no access to the internal structure, parameters, or intermediate activations of the model to explain. We summarize our main contributions as:

[1] We examine the shortcomings of current explanation

techniques for time series models through the lens of Concept Learning and Causal Counterfactual. To address these, we introduce `TimeSAE` , a framework leveraging a SAE for Concept Learning of Time Series, and mitigates distribution shift by generating samples from learned concepts aligning with the original distribution.

[2] To ensure a *faithful explanation* of `TimeSAE` 's outputs, we provide a theoretical guarantee that its structure can approximate counterfactual explanations. In particular, it can identify which dictionary atoms would lead the black-box model to produce an alternative prediction.

[3] We evaluate our method on eight time series datasets, demonstrating its superior performance to existing explanation methods. We further highlight its effectiveness in a real-world case, and made the code available in an easy-to-use library `TimeSAE` -Lib[1].

[4] We introduce *EliteLJ*, a new open-source dataset designed to benchmark time-series explanation methods. The dataset consists of skeleton-based motion data from long jump athletes and includes expert annotations labeling key phases (e.g., run-up, take-off, flight, landing) and qualitative assessments (e.g., good vs. bad take-off, correct vs. incorrect landing posture). This combination provides a realistic use case for evaluating explainability methods on sports performance data.

## 2. Related Work

**Concepts-based XAI and Sparse Autoencoders.** Concept-based Interpretable Networks (CoINs) encompasses models

---

[1] https://oublalkhalid.github.io/TimeSAE/

using human-interpretable concepts for prediction (Parekh et al., 2025) for white box models. Existing work in vision specifies concepts using either human supervision to select and provide their concept labels (Koh et al., 2020) or extracting them automatically with large language models (Oikarinen et al., 2023). Other works exploit unsupervised methods to automatically discover concepts (Alvarez Melis & Jaakkola, 2018; Sarkar et al., 2022; Oublal et al., 2024) in by-design approaches and some of them (Parekh et al., 2021) leverage sparsity and diversity constraints directly on the concept activations, which is close to the approach adopted in this paper. Unlike existing unsupervised concept learning methods, which only emphasize properties such as *faithfulness*, we also focus on *causality* and *compositionality* of concepts, i.e. concepts being presents simultaneously and composed to form complex patterns, including in time series. Following Mikolov et al. (2013) on compositional word vectors, researchers have increasingly investigated how deep learning models exhibit compositional behavior (Brady et al., 2023; Wiedemer et al., 2024).

**Counterfactual Explanations.** Counterfactual explanations can be generated with various guidances (Ye & Keogh, 2009; Bahri et al., 2024; Li et al., 2024; Tonekaboni et al., 2020; Li et al., 2023; Wang et al., 2023). However, distributional shift remains a critical issue. To address this, Liu et al. (2024c); Jang et al. (2025) generates in-domain perturbations with contrastive learning. Liu et al. (2024b) propose an information bottleneck-based objective function to ensure model faithfulness while avoiding distributional shifts. More recently, StartGrad (Uendes et al., 2025) uses an information-theoretic framework to balance masking-induced degradation with representation complexity, and ORTE (Yue et al., 2025) applies optimal information retention for the masked time-series explanations. We address this by generating samples from learned concepts aligning with the original data distribution.

**Temporal Interactions in Explanations.** Due to the specific nature of black-box model time series explanation, in both classification and regression, effects of time step must be taken into account. Leung et al. (2023) aggregate the impact across subsequent time windows while Tonekaboni et al. (2020) and Suresh et al. (2017) quantify the significance of a time step by measuring its effect on model prediction. Despite these advances, challenges remain in fully addressing the complex interactions between observational points. Yang et al. (2024) introduce time-step interactions, and Märtens & Yau (2020) propose "Variance decomposition" to quantify the variance attributed to each component. In our work, we propose a decomposition of the decoder that models and links these interactions to concepts for enhanced explanations and interpretability.

## 3. A Time Series Sparse Autoencoder for Faithful Explanations

To achieve an accurate explanation, we propose `TimeSAE`, a framework leveraging a Sparse Autoencoder that maintains the informativeness of time series for the black-box model. We show how $\mathcal{E}$ inverts the decoder $g$ for more robust concept explanations, while generating approximate CFs for more faithful explanations. At the sequence level, `TimeSAE` offers a feature-level decomposition using the temporal decoder $g$, which maps concept activations to their corresponding score activations.

**Notation.** This work focuses on explainability in time series for both regression and classification. Throughout this work, a time series instance $\mathbf{x} \in \mathcal{X} \subseteq \mathbb{R}^{D \times T}$ is represented by a $D \times T$ real-valued matrix, where $T$ is the number of time steps in the series, and $D$ is the feature dimension. If $D > 1$, the time series is multivariate; otherwise, it is univariate. We denote by $\mathcal{D} = \{(\mathbf{x}_i, \mathbf{y}_i) | i \in [N]\}$ the training set, which consists of $N$ instances $\mathbf{x}_i$ along with their associated labels $\mathbf{y}_i$, where $\mathbf{y}_i \in \mathcal{Y}$ and $\mathcal{Y}$ is the set of all possible continuous or discrete labels. A black-box time series predictor $f(\cdot)$ takes an instance $\mathbf{x} \in \mathcal{X}$ as input, and outputs a label $f(\mathbf{x}) \in \mathcal{Y}$. We further define an overcomplete sparse autoencoder $(\mathcal{E}, g)$, denoted for simplicity by SAE, where $\mathcal{E} : \mathcal{X} \to \mathcal{C}$, designed to extract concept explanations, with $\mathcal{C}$ the concept space. The activated concepts $\mathbf{c} \in \mathcal{C}$ are mapped back in the input space through the decoder $g : \mathcal{C} \to \mathbb{R}^{D \times T}$, resulting in an *explanation-embedded instance* $\widetilde{\mathbf{x}} \in \mathbb{R}^{D \times T}$. For a positive integer $n$, let $[n] = \{1, \cdots, n\}$ denote the set of integers from 1 to $n$, and we use $|\cdot|$ to represent either the dimension (or length) of a vector, or the cardinality of a set.

### 3.1. Sparse Autoencoder

The Sparse Autoencoder, defined by a pair of encoder $\mathcal{E}$ and decoder $g$ functions, decomposes each input $\mathbf{x}$ into a sparse combination of learned feature directions $\mathbf{c}$, derived from a dictionary $\mathbf{M}$. It then generates the reconstruction $\tilde{\mathbf{x}}$ of the input. This process is summarized as:

$$\mathbf{c} := \mathcal{E}(\mathbf{x}) = \sigma(\mathbf{M}\mathbf{x} + \mathbf{b}), \quad \tilde{\mathbf{x}} := g(\mathbf{c}), \qquad (1)$$

where $\mathbf{M} \in \mathbb{R}^{d \times (D \cdot T)}$, with $d := |\mathbf{c}| = r \cdot (D \cdot T)$, has its rows normalized to unit norm. This prevents scale ambiguity, i.e., the model "cheating" the sparsity objective by inflating dictionary weights to allow vanishingly small activations (Gao et al., 2024). This ensures the sparsity penalty targets feature presence rather than numerical scale. Here, $r$ is a hyperparameter controlling the size of $d$, and $\mathbf{b} \in \mathbb{R}^d$ are learned parameters. In such expression, $\mathcal{E}(\mathbf{x})$ is a sparse, non-negative vector of feature magnitudes present in the input activation. The rows of $\mathbf{M}$ represent the learned feature directions that form the dictionary used by the SAE for decomposition.

The activation function $\sigma$ varies: ReLU or gated (Rajamanoharan et al., 2024a) in some cases, while TopK SAEs (Makhzani & Frey, 2013) keep only the top-$k$ active concepts. Unfortunately, we find that this may often result in "dead" concepts similar to the phenomenon observed in LLMs (Gao et al., 2024), where some components don't actively contribute at all from the start of learning. A detailed discussion is provided in Appendix B.1.2.

To further boost fidelity, Rajamanoharan et al. (2024b) propose the JumpReLU activation, denoted as JumpReLU$_\phi$, which extends the standard ReLU-based SAE architecture (Ng, 2011) by incorporating an additional positive learnable parameter $\phi \in \mathbb{R}_+^{d \times (D \cdot T)}$ which acts as a feature-specific threshold vector. JumpReLU uses a learnable threshold $\phi_k$ for each concept feature $k$, and a feature is active only if the encoder output exceeds $\phi_k$. This relaxation prevents *"dead"* activations, stabilizing concept learning and improving reconstruction fidelity.

**Loss functions.** Within our framework `TimeSAE`, we generate the reconstructed instance $\tilde{x}$ through the JumpReLU$_\phi$ activation, which guarantees better learning of the concept dictionary and minimizes reconstruction error. Formally, the activation function is defined element-wise for any input scalar $u$ and learnable threshold $\phi$ as JumpReLU$_\phi(\mathbf{c}) :=$ $\mathbf{c} \cdot H(\mathbf{c} - \phi)$ where $\phi \in \mathbb{R}_+^d$ is the learnable threshold vector, and $H(\cdot)$ is the Heaviside step function ($H(x) = 1$ if $x > 0$, else 0). Consequently, our objective function is defined as:

$$\mathcal{L}_{\text{SAE}}(\mathbf{x}; \mathcal{E}, \boldsymbol{g}) := \underbrace{||\mathbf{x} - \boldsymbol{g}(\mathcal{E}(\mathbf{x}))||_2^2}_{\mathcal{L}_{\text{rec-fidelity}}} + \underbrace{\eta \, \mathbf{s}(\mathcal{E}(\mathbf{x}))}_{\mathcal{L}_{\text{sparsity}}}, \quad (2)$$

where $\mathbf{s}(\cdot)$ is a function of the concepts' activations that penalises the non-sparse decompositions (i.e., JumpReLU$_\phi(\mathbf{c})$) is explicitly the $L_0$ pseudo-norm of the active features: $\mathbf{s}(\mathcal{E}(\mathbf{x})) := ||\mathcal{E}(\mathbf{x})||_0 = \sum_k H(\mathbf{c}_k - \phi_k)$ and the *sparsity coefficient* $\eta$ controls the trade-off between sparsity and reconstruction fidelity.

**Temporal Concept Learning** In `TimeSAE`, the encoder and decoder use *Block Temporal Convolutional Networks (TCN)* (Bai et al., 2018) to capture temporal dependencies. The encoder maps inputs to a sparse latent space, normalized for stable training, and applies JumpReLU$_\phi$ with learnable thresholds for robust concept activations. *Squeeze-and-Excitation (SE)* blocks (Hu et al., 2018) adaptively reweight feature channels. The decoder mirrors this design, reconstructing outputs through TCN layers with normalization SE. Architectural details are provided in Appendix B.7.

### 3.2. Faithfulness via Counterfactual Explanations

To measure the causal effect of a concept on the model prediction, we rely on the *Causal Concept Effect (CaCE)*

(Goyal et al., 2019) which we formally describe as follows.

**Definition 2** (**CaCE** (Goyal et al., 2019))**.** *Given an intervention* $I_k : \boldsymbol{c}_k \mapsto \boldsymbol{c}_k'$, *a black-box model* $f : \mathcal{X} \to \mathcal{Y}$ *and a dataset* $\mathcal{D} = \{(\mathbf{x}_i, \mathbf{y}_i) | i \in [N]\}$ *of size* $N$, *the Causal Concept Effect (CaCE) is:*

$$CaCE_f(I_k) = \mathbb{E}_{\mathbf{x} \sim \mathcal{D}}\left[f(\mathbf{x}) | do(\boldsymbol{c}_k = I_k(\boldsymbol{c}_k))\right] \quad (3)$$
$$- \mathbb{E}_{\mathbf{x} \sim \mathcal{D}}\left[f(\mathbf{x}) | do(\boldsymbol{c}_k = \boldsymbol{c}_k)\right]. \quad (4)$$

The causal effect and explanation of a model are both related to counterfactuals (CFs). This enables causal estimation in a model-agnostic manner as CFs can be obtained using only the encoder $\mathcal{E}$. We can now define the Approximated Counterfactual :

**Definition 3** (**Approximated Counterfactual Explanation** (**Gat et al., 2023**))**.** *Given a dataset* $\mathcal{D} = \{(\mathbf{x}_i, \mathbf{y}_i) | i \in [N]\}$ *of size* $N$, *an encoder* $\mathcal{E} : \mathcal{X} \to \mathcal{C}$ *and an intervention* $I_k : \boldsymbol{c}_k \mapsto \boldsymbol{c}_k'$, *the approximated counterfactual explanation* $S_{cf}$ *is defined to be:*

$$S_{cf}(\mathcal{E}, I_k, \boldsymbol{c}_k, \boldsymbol{c}_k') = \frac{1}{|\mathcal{D}|} \sum_{\mathbf{x} \in \mathcal{D}} \mathcal{E}(\tilde{\mathbf{x}}_{\boldsymbol{c}_k'}) - \mathcal{E}(\tilde{\mathbf{x}}_{\boldsymbol{c}_k}), \quad (5)$$

*where* $\tilde{\mathbf{x}}_{\boldsymbol{c}_k'}$ *is the explanation-embedded instance after intervention* $I_k$, *and* $\tilde{\mathbf{x}}_{\boldsymbol{c}_k}$ *before intervention.*

In Appendix A, we provide a discussion of the Approximated CF Explanation methods. Faithfulness, as outlined in Definition 1, remains only partially addressed by the proposed explainable model. While it promotes order-preserving diversity, it does not directly address how faithful it is. In Gat et al. (2023), counterfactuals with causal relationships are shown to ensure faithfulness of explanation by validating that higher-ranked interventions have greater causal effects. Building on this, we prove the following result, showing that our explanation method preserves the relative ordering of causal effects under bounded approximation error, defined as *order-faithfulness*.

**Theorem 1** (**Faithfulness in Sparse Autoencoder-Based Approximate Counterfactuals**)**.** *Let* $\mathbf{x}$ *be a time-series input and* $f$ *a black-box model whose true output is* $\mathbf{y} = f(\mathbf{x})$. *Suppose* $(\mathcal{E}, \boldsymbol{g})$ *is an encoder-decoder, where* $\mathcal{E}$ *encodes* $\mathbf{x}$ *to latent concepts, and* $\boldsymbol{g}$ *decodes these concepts into* $\widetilde{\mathbf{x}} = \boldsymbol{g}(\mathcal{E}(\mathbf{x}))$ *such that* $\forall \mathbf{x} \in \mathcal{D}, f(\widetilde{\mathbf{x}}) \approx f(\mathbf{x})$. *For an intervention, define an* approximate counterfactual $S_{cf}$ *Definition 3 by altering concepts* $\mathbf{c} \mapsto \mathbf{c}^{cf}$, *and let* $\widetilde{\mathbf{x}}^{cf} = \boldsymbol{g}(\mathbf{c}^{cf})$. *Assume that*

$$\mathbb{E}_{\mathbf{x} \sim \mathcal{D}}\left[|f(\widetilde{\mathbf{x}}^{cf}) - \mathbf{y}^{cf}|\right] \leq \epsilon_{cf}, \quad (6)$$

*where* $\mathbf{y}^{cf}$ *is the "true" counterfactual label (i.e., what* $f(\mathbf{x})$ *would be under the exact causal intervention), and* $\epsilon_{cf}$ *is a small approximation error. Then, for any pair of*

interventions $I_1 : c_1 \mapsto c_1'$ and $I_2 : c_2 \mapsto c_2'$, *if the* true *causal effects satisfy*

$$CaCE_f(I_1, c_1, c_1') > CaCE_f(I_2, c_2, c_2'), \quad (7)$$

*there exists a sufficiently small $\epsilon_{cf}$ so that*

$$\mathbb{E}_{\mathbf{x} \sim \mathcal{D}}\left[f(\widetilde{\mathbf{x}}_{I_1}^{cf}) - f(\widetilde{\mathbf{x}})\right] > \mathbb{E}_{\mathbf{x} \sim \mathcal{D}}\left[f(\widetilde{\mathbf{x}}_{I_2}^{cf}) - f(\widetilde{\mathbf{x}})\right], \quad (8)$$

*where $\widetilde{\mathbf{x}}_{I_1}^{cf}$ and $\widetilde{\mathbf{x}}_{I_2}^{cf}$ are the explanation-embedded instances respectively obtained after interventions $I_1$ and $I_2$. This preserves the* ordering *of causal effects, i.e. order faithfulness.*

*Proof Sketch.* We define the actual and approximate causal effects of interventions and limit approximation errors. By ensuring that the combined approximation and reconstruction errors are smaller than the actual causal effects, we guarantee that order is preserved. Thus, approximate counterfactuals based on a sparse autoencoder preserve the causal effects' order. The full proof, along with empirical validation through experiments, is provided in Section A.

**Practical Implementation.** To enforce faithfulness, we use a contrastive loss InfoNCE (Oord et al., 2018) during the training of the Sparse Autoencoder. We define the positive pairs as counterfactuals from the same intervention $I_k$, while negative pairs come from different interventions $I_j$, ensuring $CaCE_f(I_k) > CaCE_f(I_j)$. The contrastive loss is formulated as:

$$\mathcal{L}_{cf} = -\log\left(\frac{\exp(\text{sim}(f(\widetilde{\mathbf{x}}_{I_k}^{cf}), f(\widetilde{\mathbf{x}'}_{I_k}^{cf}))/\tau)}{\sum_j \exp(\text{sim}(f(\widetilde{\mathbf{x}}_{I_k}^{cf}), f(\widetilde{\mathbf{x}}_{I_j}^{cf}))/\tau)}\right), \quad (9)$$

where $\text{sim}(\cdot,\cdot)$ is the cosine similarity and $\tau$ is a temperature parameter that controls the sharpness of the similarity distribution. In practice, following Algorithm 1, the counterfactual can be obtained in an unsupervised manner, similarly to (Yan & Wang, 2023).

### 3.3. Generalization in Out-of-Distribution

For a generator $\boldsymbol{g}$ that is consistent for an OOD sample, Wiedemer et al. (2024) showed that an autoencoder will *slot-identify* concepts $\mathbf{c}$ in $\mathcal{C}$. The conditions on the encoder discussed in the previous section aim to ensure that $\mathcal{E}$ inverts $\boldsymbol{g}$ over the entire input space $\mathcal{X}$. This behavior is encouraged within the training space $\mathcal{X}$ (in-distribution) by minimizing the *reconstruction fidelity* objective $\mathcal{L}_{\text{rec-fidelity}}$. However, no mechanism is in place to enforce this inversion behavior of $\mathcal{E}$ on $\boldsymbol{g}$ also outside of $\mathcal{X}$ (out-of-distribution). To address this, we propose to use the following compositional consistency loss (Wiedemer et al., 2024):

$$\mathcal{L}_{cc}(\mathcal{E}, \boldsymbol{g}, \mathcal{C}') := \mathbb{E}_{\mathbf{c}' \sim q_{\mathbf{c}'}}\left[\left\|\mathcal{E}(\boldsymbol{g}(\mathbf{c}')) - \mathbf{c}'\right\|_2^2\right], \quad (10)$$

where $q_{\mathbf{c}'}$ is a distribution with support $\mathcal{C}'$. The loss can be viewed as sampling an OOD combination of concepts $\mathbf{c}'$ by composing inferred in-distribution concepts (i.e. from $\mathcal{E}(\mathbf{x}), \forall \mathbf{x} \in \mathcal{X}$). This synthesized concept is then passed through the decoder to generate an OOD input $\boldsymbol{g}(\mathbf{c}')$. This sample is then re-encoded with $\mathbf{c}' = \mathcal{E}(\boldsymbol{g}(\mathbf{c}'))$. Finally, the loss regularizes the encoder $\mathcal{E}$ to serve as an approximate inverse of the decoder function for OOD samples. The choice of the compositionality of concepts is crucial and has proved to be stable (Miao et al., 2022).

### 3.4. Concept Activation Alignment and Discovery

Understanding which part of the input time series, i.e., the global features, depends on learned concepts and how these concepts interact is a core challenge in time-series interpretation. This insight is essential for building human trust and enabling practical applications. We propose a novel approach that formalizes the interpretation of global features using a decompositional decoder structure and probabilistic sparsity masks.

**Automated Global Features Interpretation.** Our method introduces Bernoulli random variables introduce $\boldsymbol{m}_k^{(j)} \sim$ Bernoulli($\boldsymbol{p}_0$) to probabilistically model the presence or absence of concept dependence for each interaction order $k \in \{1, \ldots, d\}$, and for each concepts $j \in \{1, \ldots, d\}$, where each $\boldsymbol{m}_k^{(j)}$ indicates the presence of a non-zero effect of $\boldsymbol{\psi}_k$ and thus of the concepts. More formally, $\boldsymbol{g}(\mathbf{c})$ follows:

$$\boldsymbol{g}(\mathbf{c}) := \boldsymbol{\psi}_0 + \sum_{j=1}^{d} \boldsymbol{\psi}_1(\boldsymbol{c}_j) + \sum_{j=1}^{d-1} \boldsymbol{\psi}_2(\boldsymbol{c}_j, \boldsymbol{c}_{j+1}) \quad (11)$$

$$+ \sum_{j=1}^{d-2} \boldsymbol{\psi}_3(\boldsymbol{c}_j, \boldsymbol{c}_{j+1}, \boldsymbol{c}_{j+2}) + \cdots + \boldsymbol{\psi}_d(\mathbf{c}),$$

where $\boldsymbol{\psi}_0$ denote the bias term, $\boldsymbol{\psi}_1$ captures the first-order contribution of $\boldsymbol{c}_j$, and $\boldsymbol{\psi}_2$ models the second-order interaction between $\boldsymbol{c}_j$ and $\boldsymbol{c}_{j-1}$, with summation over terms $\boldsymbol{\psi}_k$ for $k = 2, \ldots d$. Higher-order terms $\boldsymbol{\psi}_d$ represent interactions across successive concepts $\mathbf{c}$. These terms are elementwise multiplied with their respective Bernoulli sparsity masks $\boldsymbol{m}_k^{(j)}$, activating or deactivating specific interaction effects. This generalizes the neural functional ANOVA decomposition (Märtens & Yau, 2020) and GAM (Yang et al., 2024), offering interpretability by visualizing individual concept impacts and their interactions.

**Concept Alignment.** To enhance interpretability, particularly for human users, we need a mechanism to link the model's internal concepts to human-understandable abstractions with fewer labels. Following training, our framework discovers and aligns learned concepts using techniques inspired by Crabbé & van der Schaar (2022) works, Concept

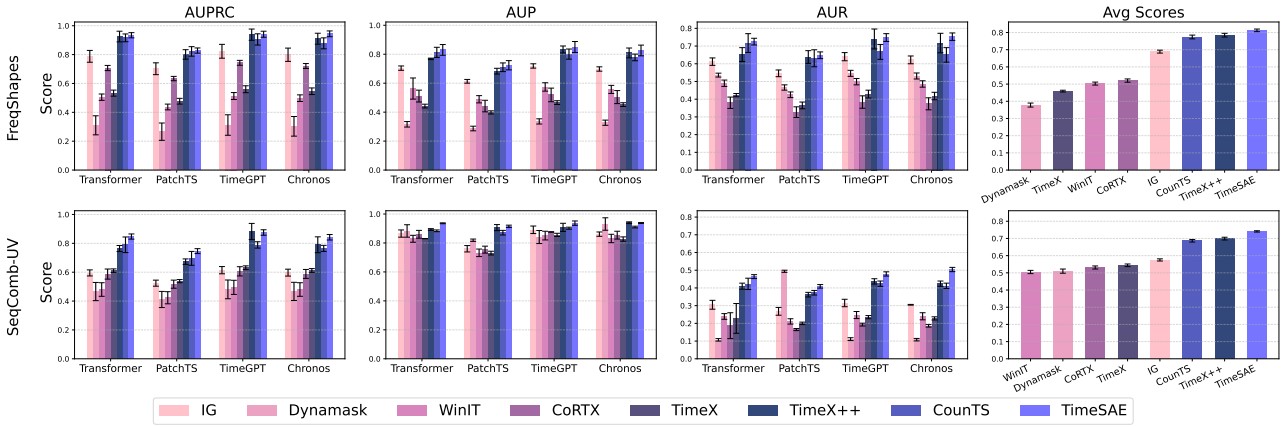

*Figure 2.* Explanation performance on all datasets and metrics (AUPRC, AUP, AUR). Higher is better. The rightmost panel shows average scores. Methods are ranked left to right from worst to best.

Activation Regions (CAR), better suited to time series tasks as it doesn't assume linear separability. Specifically, we enable the manual definition of low-level concepts, which are subsequently aligned with higher-level abstractions learned by the model. This alignment is performed in a supervised manner using kernel-based Support Vector Regression (SVR) and Support Vector Classification (SVC) to distinguish between the activations produced by the network for samples containing the target concept and those for randomly selected samples. We provide the Algorithm in the Appendix B.9.1.

**Training Setting.** `TimeSAE` minimizes all the previously described loss functions. Notably, we include a *reconstruction term* to ensure *label fidelity* between the prediction made directly from $f(\mathbf{x})$ and the one obtained from the reconstructed input $f\big(g(\mathcal{E}(\mathbf{x}))\big)$. We refer to this term as the *label-fidelity loss*, denoted by $\mathcal{L}_{\text{label-fidelity}} := ||f(\mathbf{x}) - f\big(g(\mathcal{E}(\mathbf{x}))\big)||_2^2$, and the overall loss is

$$\mathcal{L}_{\text{TIMESAE}} = \mathcal{L}_{\text{SAE}} + \mathcal{L}_{\text{label-fidelity}} + \alpha\mathcal{L}_{cc} + \lambda\mathcal{L}_{cf}, \quad (12)$$

where $(\alpha, \lambda) \in \mathbb{R}^2$ are hyperparameters adjusting the losses, and `TimeSAE` is optimized end-to-end.

## 4. Experimental Setup

**Black-Box Models.** We employ two types of black-box models: we trained a Transformer-based classifier (Vaswani et al., 2017), DLinear (Zeng et al., 2023), and PatchTS (Nie et al., 2022) models for regression tasks, with hyperparameters carefully tuned to maximize predictive performance. In addition, we incorporate pretrained large-scale models for forecasting and classification: TimeGPT (Garza et al., 2023), accessed via API; Chronos (Ansari et al., 2024), an open-source black-box model with 188 billion parameters designed for regression tasks. Due to space constraints, additional models such as Moments (Goswami et al., 2024),

TimeFM (Das et al., 2024), and Informer (Zhou et al., 2021) are reported in the Appendix B.3.1. For all predictors, we verified that the models achieved satisfactory performance on the testing set before the explainability evaluation.

**Datasets.** We use two synthetic datasets with known ground-truth explanations: **(1) FreqShapes** and **(2) SeqComb-UV** adapted from Queen et al. (2023). Datasets are designed to capture diverse temporal dynamics in both univariate and multivariate settings (we give more details in Section B.1.1). We employ four datasets from real-world time series. For classification tasks, we consider **(1) ECG** dataset (Moody & Mark, 2001) that consists of arrhythmia detection, which includes cardiac disorders with ground-truth explanations defined as the QRS interval (Queen et al., 2023). **(2) PAM** (Reiss & Stricker, 2012) human activity recognition. For regression, **(3) ETTH-1** and **(4) ETTH-2** which are energy demand datasets (Ruhnau et al., 2019). Finally, we introduce **(5) EliteLJ Dataset**, a new real-world sports dataset, consisting of skeletal athlete pose sequences along with the corresponding performance metrics. More details are provided in the Appendix B.1.2.

**Explanation Evaluation.** Given that precise salient features are known, we utilize them as ground truth for evaluating explanations. At each time step, features causing prediction label changes are attributed an explanation of 1, whereas those that do not affect such changes are 0. Following Crabbé & Van Der Schaar (2021), we evaluate the quality of explanations with area under precision (AUP), area under recall (AUR), and also AUPRC, for consistency from Queen et al. (2023), which combines the two. Following TimeX (Liu et al., 2024b), we assess distributional similarity using KL divergence and MMD (Gretton et al., 2012), with smaller values denoting closer alignment. We further estimate the *log-likelihood* of explanations via KDE (Parzen, 1962).

**Faithfulness Evaluation.** We evaluate *faithfulness* to assess

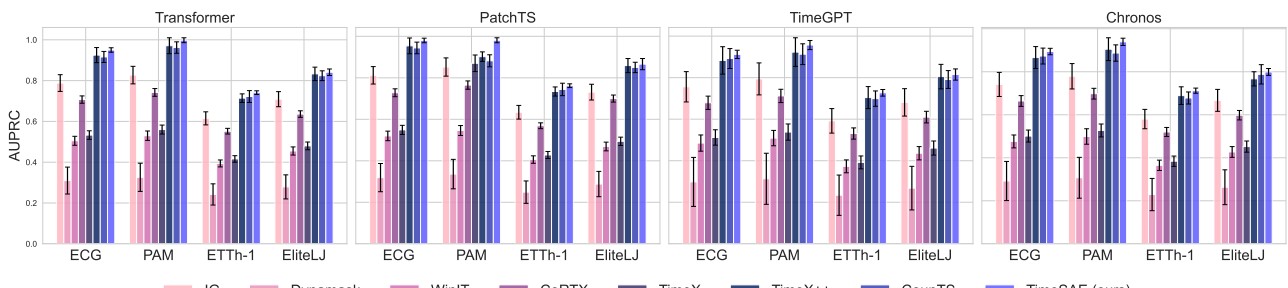

*Figure 3.* AUPRC explanation performance (higher is better) across methods for each dataset.

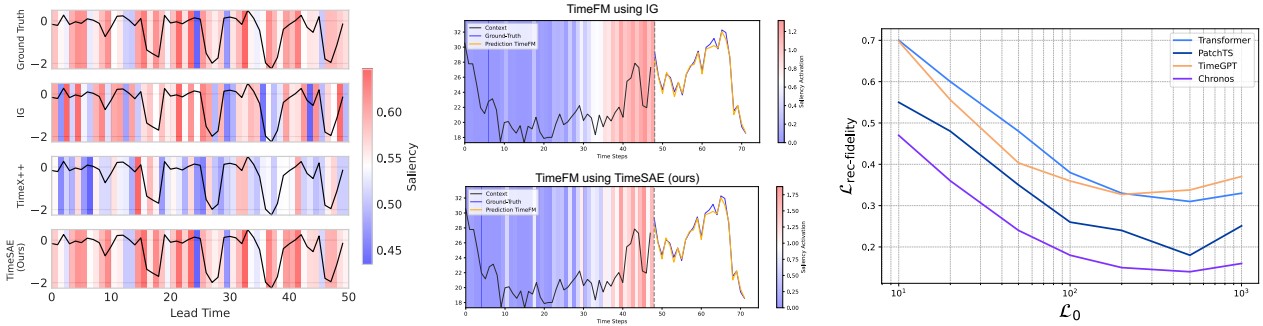

*Figure 4. Left:* Examples of explanations compared to ground truth on the FreqShapes dataset. *Middle:* A 24-hour forecast based on a 48-hour history from the ETTh1 dataset. The heatmaps visualize model explanations generated by TimeSAE compared to IG for interpreting the TimeFM foundation model. IG tends to attribute importance broadly across many lagged events, whereas TimeSAE focuses on the most relevant temporal regions, producing more faithful explanations. *Right:* Effect of sparsity on reconstruction fidelity. Increasing sparsity generally improves interpretability; however, excessive sparsity can reduce fidelity.

the reliability of our interpretations, following the methodology introduced in Parekh et al. (2022). Faithfulness measures whether the features identified as relevant are truly important for the classifier's predictions. Since "ground-truth" feature importance is rarely available, attribution-based methods typically evaluate faithfulness by removing features (e.g., setting their values to zero) and observing changes in the classifier's output. However, we enable the simulation of feature removal in time-series data by deactivating a set of components. For a sample $\mathbf{x}$ with predicted $\mathbf{y}$, we remove the set of relevant components in $\mathbf{c}$ to obtain $\mathbf{c}^-$ and generate a new instance $\widetilde{\mathbf{x}}^- = g(\mathbf{c}^-)$. The faithfulness score is then computed as: $\mathcal{F}_{\mathbf{x}} = ||f(\mathbf{x}) - f(\widetilde{\mathbf{x}}^-)||_2^2$ where $f(\mathbf{x})$ and $f(\widetilde{\mathbf{x}}^-)$ represent the model's output before and after concept removal. A higher $\mathcal{F}_{\mathbf{x}}$ indicates a greater impact of the removed components on the output, supporting the faithfulness of the interpretation.

**Baselines details.** The proposed method TimeSAE is evaluated against eight state-of-the-art explainability methods: Integrated Gradients (IG) (Sundararajan et al., 2017), Dynamask (Crabbé & Van Der Schaar, 2021), WinIT (Leung et al., 2023), CoRTX (Chuang et al., 2023), SGT-GRAD (Ismail et al., 2021), TIMEX (Queen et al., 2023), TIMEX++ (Liu et al., 2024b), and CounTS (Yan & Wang, 2023), as well as the more recent methods StartGrad Uendes et al. (2025), TIMING (Jang et al., 2025) and ORTE (Yue

et al., 2025) based optimal information retention to find explanation for time series. Further implementation details are provided in the Appendix B.6.

### 4.1. Synthetic and Real-world Datasets

Figure 2 and Figure 3 report the performance of different explanation methods on univariate and multivariate datasets, presented as the mean and standard deviation over 10 random seeds (applies to all subsequent tables/figures). Our approach consistently achieves the best results across metrics (AUPRC, AUP, and AUR).

The performance metric (AUPRC) on real datasets, illustrated in Figure 3, show that TimeSAE surpasses existing methods, including TimeX++, across all datasets. We perform paired t-tests at the 5% significance level to evaluate whether TimeSAE significantly outperforms other methods. The results indicate that TimeSAE achieves statistically the highest performance on several datasets (e.g., ECG and PAM), while other models like TimeX++ and CounTS remain competitive.

### 4.2. Ablation study

**Concepts Consistency in OOD** To assess the effectiveness of the Concepts Consistency in out-of-distribution (OOD) generalization, we evaluate the model's ability to generate explanations that remain faithful and robust when exposed

*Table 1.* **Generalization across diverse distribution shifts.** We evaluate explanation quality under both in-distribution (ID) and multiple out-of-distribution (OOD) settings spanning temporal-resolution shift, cross-domain transfer, cross-dataset HAR transfer, and athlete-population shift. Higher is better for KDE, AUPRC, and $\mathcal{F}_\mathbf{x}$; lower is better for KL and MMD. TimeSAE consistently achieves the strongest OOD robustness and exhibits a substantially smaller ID→OOD degradation than prior methods. The colors represent the top Top-1 , Top-2 , and Top-3 rankings.

| Dataset Domain | Method | Generalization | KDE↑ | KL↓ | MMD↓ | AUPRC↑ | $\mathcal{F}_\mathbf{x}$ ↑ |
|---|---|---|---|---|---|---|---|
| ● ETTh1→ETTh1-val | IG | ID | -46.27±1.4 | 0.301±0.027 | 0.029±0.005 | 0.418±0.047 | 1.36±0.07 |
| | TimeX | ID | -44.31±1.2 | 0.204±0.022 | 0.020±0.003 | 0.702±0.05 | 1.71±0.08 |
| | TimeX++ | ID | -44.05±1.1 | 0.196±0.019 | 0.018±0.002 | 0.719±0.05 | 1.78±0.07 |
| | TimeSAE | ID | **-43.55±1.1** | **0.182±0.010** | **0.016±0.002** | **0.741±0.05** | **2.12±0.05** |
| ○ ETTh1→ETTh2 | IG | OOD | -49.62±1.7 | 0.362±0.034 | 0.124±0.013 | 0.388±0.032 | 1.28±0.08 |
| | TimeX | OOD | -48.91±1.4 | 0.272±0.031 | 0.104±0.011 | 0.611±0.041 | 1.66±0.08 |
| | TimeX++ | OOD | -48.70±1.3 | 0.261±0.028 | 0.103±0.010 | 0.628±0.04 | 1.73±0.07 |
| | TimeSAE | OOD | **-47.21±1.3** | **0.245±0.020** | **0.089±0.010** | **0.641±0.030** | **2.09±0.06** |
| ○ ETTh1→ETTm1 | IG | Energy | -0.412±0.03 | 0.538±0.04 | 0.227±0.02 | 0.371±0.034 | 1.24±0.08 |
| | TimeX | Energy | -0.348±0.02 | 0.471±0.03 | 0.192±0.02 | 0.589±0.035 | 1.79±0.07 |
| | TimeX++ | Energy | -0.339±0.02 | 0.463±0.03 | 0.184±0.01 | 0.601±0.034 | 1.86±0.07 |
| | TimeSAE | Energy | **-0.321±0.02** | **0.448±0.03** | **0.178±0.01** | **0.619±0.033** | **2.01±0.07** |
| ○ Weather→ETTh1 | IG | Weather→Energy | -0.471±0.04 | 0.604±0.05 | 0.251±0.02 | 0.352±0.036 | 1.19±0.09 |
| | TimeX | Weather→Energy | -0.401±0.03 | 0.532±0.04 | 0.208±0.02 | 0.572±0.037 | 1.74±0.08 |
| | TimeX++ | Weather→Energy | -0.392±0.03 | 0.524±0.04 | 0.202±0.02 | 0.583±0.036 | 1.81±0.08 |
| | TimeSAE | Weather→Energy | **-0.378±0.03** | **0.512±0.04** | **0.198±0.02** | **0.598±0.035** | **1.95±0.08** |
| ○ PAMAP2→OPPORTUNITY | IG | Activity (HAR) | -0.489±0.04 | 0.621±0.05 | 0.266±0.02 | 0.361±0.035 | 1.22±0.09 |
| | TimeX | Activity (HAR) | -0.416±0.03 | 0.548±0.04 | 0.221±0.02 | 0.591±0.036 | 1.77±0.08 |
| | TimeX++ | Activity (HAR) | -0.409±0.03 | 0.541±0.04 | 0.217±0.02 | 0.602±0.035 | 1.84±0.07 |
| | TimeSAE | Activity (HAR) | **-0.398±0.03** | **0.531±0.04** | **0.211±0.02** | **0.612±0.034** | **1.98±0.07** |
| ○ EliteLJ (elite→interm.) | IG | Sports | -0.358±0.03 | 0.481±0.04 | 0.189±0.02 | 0.411±0.04 | 1.31±0.08 |
| | TimeX | Sports | -0.286±0.02 | 0.401±0.03 | 0.152±0.01 | 0.648±0.037 | 1.88±0.07 |
| | TimeX++ | Sports | -0.279±0.02 | 0.394±0.03 | 0.148±0.01 | 0.659±0.037 | 1.93±0.07 |
| | TimeSAE | Sports | **-0.267±0.02** | **0.389±0.03** | **0.143±0.01** | **0.671±0.038** | **2.05±0.06** |

*Table 2.* The Faithfulness $\mathcal{F}_\mathbf{x}$ metric performance across classification ($^\dagger$) and regression ($^\ddagger$) tasks with different datasets. Higher values are better, and the colors represent the top Top-1 , Top-2 , and Top-3 rankings.

| Black-Box → | Transformer | | PatchTS | | DLinear | |
|---|---|---|---|---|---|---|
| Method ↓ | ECG$^\dagger$ | PAM$^\dagger$ | ETTH-1$^\ddagger$ | ETTH-2$^\ddagger$ | EliteLJ$^\ddagger$ | Rank |
| IG | 0.92±0.102 | 0.89±0.104 | 1.00±0.107 | 0.95±0.101 | 0.91±0.109 | 9.0 |
| Dynamask | 1.05±0.099 | 1.00±0.095 | 1.15±0.096 | 1.12±0.093 | 1.04±0.097 | 8.0 |
| WinIT | 1.10±0.095 | 1.08±0.092 | 1.18±0.091 | 1.17±0.090 | 1.06±0.093 | 6.9 |
| CoRTX | 1.15±0.088 | 1.10±0.087 | 1.22±0.090 | 1.20±0.088 | 1.14±0.089 | 5.6 |
| TimeX | 1.10±0.083 | 1.22±0.091 | 1.35±0.090 | 1.28±0.089 | 1.10±0.094 | 5.5 |
| ORTE | 1.51±0.009 | 1.55±0.078 | 1.71±0.077 | 1.50±0.075 | 1.43±0.072 | 4.1 |
| TIMING | 1.67±0.050 | 1.65±0.070 | 1.82±0.060 | 1.69±0.055 | 1.55±0.065 | 3.5 |
| TimeX++ | 1.65±0.097 | 1.58±0.088 | 1.75±0.084 | 1.70±0.086 | 1.44±0.087 | 3.1 |
| StartGrad | 1.68±0.010 | 1.72±0.087 | 1.90±0.085 | 1.67±0.083 | 1.65±0.080 | 2.4 |
| CounTS | **1.86±0.075** | 2.05±0.074 | 1.60±0.080 | 1.50±0.081 | **1.89±0.071** | 2.3 |
| TimeSAE (ours) | 1.78±0.078 | **2.15±0.080** | **2.12±0.072** | **2.09±0.069** | 1.86±0.032 | **1.7** |

to OOD data, i.e., tested on ETTh-2 while TimeSAE is trained on ETTh-1, widely used as OOD generalization benchmark. These datasets are collected from different countries, exhibit distinct seasonal and frequency patterns, and thus serve as a suitable testbed for OOD evaluation (Liu et al., 2024a). ID and OOD statistics are provided in Section B.14. We report in Table 1 the combined results of KDE, KL, MMD, AUPRC and $\mathcal{F}_\mathbf{x}$, comparing in-domain and cross-domain settings. These results show that our proposed TimeSAE not only achieves higher predictive performance (AUPRC) but also produces more faithful and distributionally aligned explanations (lower KDE shift, KL divergence, and MMD). To confirm that these gains, we further evaluate four distinct shifts: a temporal-resolution change, a cross-domain transfer shift. As reported in Table 1, TimeSAE consistently outperforms IG and TimeX++ across all settings and exhibits a markedly smaller ID→OOD degradation, confirming the consistency term $\mathcal{L}_{cc}$ as the main driver of out-of-distribution robustness.

**Effectiveness of Counterfactual for Faithful Explanations** We evaluate the contribution of counterfactual supervision in enhancing explanation faithfulness by evaluating the proposed framework using the $\mathcal{F}_\mathbf{x}$ score across five diverse time series datasets and multiple black-box predictors. As shown in Table 3, we examine three variants of the TimeSAE model: one trained with the full objective Equation (12), another without the counterfactual loss term $\mathcal{L}_{cf}$, and a third version trained without the consistency loss $\mathcal{L}_{cc}$. The results consistently show that incorporating counterfactual objectives significantly improves faithfulness across all settings. Notably, models trained with $\mathcal{L}_{cf}$, especially when combined with $\mathcal{L}_{cc}$, yield higher $\mathcal{F}_\mathbf{x}$ scores, demonstrating that counterfactual signals help guide the model toward learning more faithful and semantically meaningful explanations. These improvements hold across datasets and predictor types, showing the generalizability of our approach. Qualitative comparisons in Figure 4-a of TimeSAE 's explanations with the ground truth against TimeX++ and IG show that TimeSAE better captures temporal attributions.

*Table 3.* **Leave-one-out ablation of the `TimeSAE` objective** (Equation (12)) on FreqShapes, over 10 seeds. $\Delta\%$ columns report the relative change w.r.t. the full objective. The colors represent the top Top-1 , Top-2 , and Top-3 rankings. All four terms are necessary and non-redundant.

| Variant | AUPRC↑ | AUP↑ | AUR↑ | $\mathcal{F}_{\mathbf{x}}$ ↑ | $\triangle$AUPRC % | $\triangle$AUP % | $\triangle$AUR % | $\triangle\mathcal{F}_{\mathbf{x}}$ % |
|---|---|---|---|---|---|---|---|---|
| $\mathcal{L}_{\text{SAE}}$ only (vanilla SAE) | 0.521±0.045 | 0.498±0.041 | 0.476±0.038 | 1.21±0.089 | -29.7% | -30.1% | -30.9% | -42.9% |
| $\mathcal{L}_{cf}$ only | 0.501±0.049 | 0.483±0.046 | 0.461±0.043 | 1.19±0.092 | -32.4% | -32.2% | -33.1% | -43.9% |
| $\mathcal{L}_{cc}$ only | 0.489±0.051 | 0.471±0.048 | 0.452±0.045 | 1.12±0.095 | -34.0% | -33.8% | -34.4% | -47.2% |
| $\mathcal{L}_{\text{label-fid.}}$ only | 0.598±0.041 | 0.572±0.038 | 0.543±0.035 | 1.38±0.082 | -19.3% | -19.7% | -21.2% | -34.9% |
| w/o $\mathcal{L}_{cf}$ | 0.638±0.041 | 0.612±0.038 | 0.581±0.036 | 1.91±0.071 | -13.9% | -14.1% | -15.7% | -9.9% |
| w/o $\mathcal{L}_{cc}$ | 0.695±0.036 | 0.671±0.033 | 0.638±0.031 | 1.61±0.076 | -6.2% | -5.8% | -7.4% | -24.1% |
| w/o $\mathcal{L}_{\text{label-fid.}}$ | 0.531±0.047 | 0.512±0.044 | 0.489±0.041 | 1.31±0.088 | -28.3% | -28.1% | -29.0% | -38.2% |
| Full objective | **0.741±0.050** | **0.712±0.047** | **0.689±0.044** | **2.12±0.050** | – | – | – | – |

**Leave-one-out ablation of the objective** To assess whether each term of Equation (12) is necessary rather than redundant, we conduct a complete leave-one-out study on FreqShapes (Table 3). Removing the counterfactual term $\mathcal{L}_{cf}$ reduces AUPRC by up to 13.9%, removing the consistency term $\mathcal{L}_{cc}$ reduces faithfulness $\mathcal{F}_{\mathbf{x}}$ by up to 24.1%, and removing the label-fidelity term yields the most severe degradation (up to 38.2%). No single term suffices on its own, confirming that the four terms are complementary, each tied to a necessary condition of Theorem 1.

**Trade-off Between Sparsity and Reconstruction** Although sparsity can improve the monosemanticity of concepts (Pach et al., 2025), forcing sparsity also leads to *lower fidelity* of reconstruction. In Figure 4-c, we evaluate reconstruction error as sparsity increases (measured by $\mathcal{L}_0$). The results indicate that when representations become overly sparse (low $\mathcal{L}_0$), reconstruction fidelity decreases (higher $\mathcal{L}_{\text{rec-fidelity}}$). We extend this analysis to the TopK sparsity mechanism (Section 3), referred to as TimeSAE-TopK; detailed results are shown in Appendix B.15. Overall, we find that `TimeSAE` achieves strong sparsity without compromising reconstruction quality. Additional results on other datasets are provided in Appendix B. These observations underscore JumpReLU's adaptability to different data distributions, whereas TopK is more sensitive to hyperparameter settings.

## 5. Conclusion and Future Work

We address the challenge of post-hoc explainability for time-series black-box models by employing sparse autoencoders and exploring the causal relationships between concepts within explanation-embedded instances to ensure faithful explanations. Our proposed approach, `TimeSAE`, automates dictionary learning, generating time series explanations that preserve labels and are consistent with the original data distribution. Through experiments on both synthetic and real-world datasets, we show that `TimeSAE` outperforms current explainability methods in terms of faithfulness and robustness to distribution shift, with successful applications in fields such as energy forecasting and sports analytics. In particular, our results highlight `TimeSAE`'s ability to accurately capture the complex dynamics of pre-trained time series models. This work emphasizes the importance of concept composition and causality in producing high-quality explanations. Future investigations could focus on constructing white-box models based on these concepts and exploring the interpretability of layers within black-box models.

**From Mechanistic Interpretability to Discovery.** Sparse dictionary learning underlies mechanistic interpretability (Olah et al., 2020; Huben et al., 2024), where features are often treated as interpretable by virtue of sparsity. However, sparsity is a regularizer, not an explanation: the directions recovered by sparse autoencoders are correlational, with no guarantee that intervening on them produces predictable or *actionable* changes in model behavior. Explanatory force instead requires a causal commitment. The CaCE objective $\mathcal{L}_{cf}$ and the order-faithfulness guarantee of Theorem 1 make this precise: intervening on a higher-CaCE concept yields a larger counterfactual effect than a lower one, enabling actionable predictions of model responses under suppression or amplification of concepts rather than merely describing what the model encodes. Causal Mechanistic Interpretability (CMI) reframes mechanistic interpretability in terms of causal interventions on learned features, replacing correlational notions with intervention-grounded semantics that support *actionable counterfactual prediction* of model behavior. This perspective extends naturally to probing, circuit analysis, and attribution patching when equipped with an explicit causal criterion, and links causally ordered concepts to the hand-designed libraries used in equation-discovery methods (Brunton et al., 2016; Pervez et al., 2024).

**Limitations** While `TimeSAE` produces faithful and interpretable explanations, it relies on sufficiently large and representative datasets, which may limit its use in data-scarce settings. Its performance is also sensitive to hyperparameters such as sparsity and dictionary size, requiring careful tuning for robustness and generalization. A full discussion is provided in Appendix C.

## Impact Statement

The `TimeSAE` improves interpretability for time series models, including large-scale foundation models. By providing high-fidelity, concept-based explanations, it significantly enhances transparency in critical domains such as healthcare and energy. Overall, `TimeSAE` supports more reliable and accountable AI, enabling human-in-the-loop verification and fostering broader public trust in automated temporal decision-making, without introducing any ethical risks.

## Acknowledgments

This work was partially funded by the ERC (Grant 853489 – DEXIM) and the Alfried Krupp von Bohlen und Halbach Foundation, which we gratefully acknowledge for their support. This work was also supported in part by Google research credits, Hi! PARIS, and the ANR/France 2030 program (ANR-23-IACL-0005).

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

# Supplementary Material

We begin by summarizing the notation used throughout the paper to improve readability and provide a consistent reference for the mathematical symbols and objects used in both the main paper and the appendix.

| Symbol | Description |
|---:|---|
| $C$ | Number of features for time series |
| $T$ | Denote the sequence length |
| $d_x$ | The dimension $C \times T$, where $T$ is time and $C$ features. |
| $d_z$ | The dimension $r \cdot C \times T$, where $r$ is the latent Dimensionality Ratio. |
| $\mathbf{x} \in \mathbb{R}^{d_x}$ | Observation |
| $\mathbf{x}^{cf} \in \mathbb{R}^{d_x}$ | The counterfactual Observation |
| $\mathbf{y} \in \mathbb{R}^{d_y}$ | Ground-truth |
| $\mathbf{y}^{cf} \in \mathbb{R}^{d_y}$ | The counterfactual ground-truth |
| $\boldsymbol{c} \in \mathbb{R}^{d_z}$ | Vector of concepts factors |
| $\mathcal{C} \subseteq \mathbb{R}^{d_z}$ | Support of concepts $\boldsymbol{c}$ |
| $I_k$ | Intervention on the concepts $\boldsymbol{c}_k$ |
| $\mathcal{E}$ | Explainer or encoder function |
| $g$ | Decoder function |
| $f$ | The Blackbox model to explain |
| $|\mathcal{D}|$ | The cardinal of the dataset $\mathcal{D}$ |

*Table 4.* Summary of the notation introduced in the TimeSAE framework.

## A. Definitions, Assumptions, and Proofs

This section provides the formal definitions, assumptions, and theoretical justifications underpinning the methods discussed in the main text. We introduce key concepts related to counterfactual explanations, specifically the Causal Concept Effect (CaCE) and its approximation in a model-agnostic setting using explainers Section A.2. We then present a formal proof of faithfulness for the proposed Approximate Counterfactual Explanation method, particularly within the framework of Sparse Autoencoders.

### A.1. Approximated Counterfactual Explanation

Causal effects and model explanations are naturally linked to counterfactuals (CFs), as they quantify how model outputs change under hypothetical interventions. However, computing exact counterfactuals often requires full knowledge of the underlying data-generating process, which is typically unavailable. To address this, we adopt an *approximated* counterfactual approach, leveraging only the explainer $\mathcal{E}$ and observed data.

**Definition 2** (CaCE (Goyal et al., 2019)). *Given an intervention* $I_k : \boldsymbol{c}_k \mapsto \boldsymbol{c}'_k$, *a black-box model* $f : \mathcal{X} \to \mathcal{Y}$ *and a dataset* $\mathcal{D} = \{(\mathbf{x}_i, \mathbf{y}_i) | i \in [N]\}$ *of size* $N$, *the Causal Concept Effect (CaCE) is:*

$$CaCE_f(I_k) = \mathbb{E}_{\mathbf{x} \sim \mathcal{D}} \left[ f(\mathbf{x}) | do(\boldsymbol{c}_k = I_k(\boldsymbol{c}_k)) \right] \tag{3}$$

$$- \mathbb{E}_{\mathbf{x} \sim \mathcal{D}} \left[ f(\mathbf{x}) | do(\boldsymbol{c}_k = \boldsymbol{c}_k) \right]. \tag{4}$$

By using approximated counterfactuals, we can efficiently estimate the causal effect of concepts on model predictions in a model-agnostic way. This allows us to generate explanations that capture the influence of individual concepts while remaining computationally tractable, even in complex, high-dimensional datasets.

**Definition 3** (**Approximated Counterfactual Explanation** (Gat et al., 2023)). *Given a dataset* $\mathcal{D} = \{(\mathbf{x}_i, \mathbf{y}_i) | i \in [N]\}$ *of size* $N$, *an encoder* $\mathcal{E} : \mathcal{X} \to \mathcal{C}$ *and an intervention* $I_k : \boldsymbol{c}_k \mapsto \boldsymbol{c}'_k$, *the approximated counterfactual explanation* $S_{cf}$ *is defined to be:*

$$S_{cf}(\mathcal{E}, I_k, \boldsymbol{c}_k, \boldsymbol{c}'_k) = \frac{1}{|\mathcal{D}|} \sum_{\mathbf{x} \in \mathcal{D}} \mathcal{E}(\tilde{\mathbf{x}}_{\boldsymbol{c}'_k}) - \mathcal{E}(\tilde{\mathbf{x}}_{\boldsymbol{c}_k}), \tag{5}$$

*where $\tilde{\mathbf{x}}_{\mathbf{c}'_k}$ is the explanation-embedded instance after intervention $I_k$, and $\tilde{\mathbf{x}}_{\mathbf{c}_k}$ before intervention.*

## A.2. Proof of Faithfulness for Approximate Counterfactuals in Sparse Autoencoders

**Theorem 1** (**Faithfulness in Sparse Autoencoder-Based Approximate Counterfactuals**). *Let $\mathbf{x}$ be a time-series input and $f$ a black-box model whose true output is $\mathbf{y} = f(\mathbf{x})$. Suppose $(\mathcal{E}, \boldsymbol{g})$ is an encoder-decoder, where $\mathcal{E}$ encodes $\mathbf{x}$ to latent concepts, and $\boldsymbol{g}$ decodes these concepts into $\widetilde{\mathbf{x}} = \boldsymbol{g}(\mathcal{E}(\mathbf{x}))$ such that $\forall \mathbf{x} \in \mathcal{D}$, $f(\widetilde{\mathbf{x}}) \approx f(\mathbf{x})$. For an intervention, define an* approximate counterfactual $S_{cf}$ *Definition 3 by altering concepts $\mathbf{c} \mapsto \mathbf{c}^{cf}$, and let $\widetilde{\mathbf{x}}^{cf} = \boldsymbol{g}(\mathbf{c}^{cf})$. Assume that*

$$\mathbb{E}_{\mathbf{x}\sim\mathcal{D}}\big[\,|\,f(\widetilde{\mathbf{x}}^{cf}) - \mathbf{y}^{cf}\,|\,\big] \;\leq\; \epsilon_{cf}, \tag{6}$$

*where $\mathbf{y}^{cf}$ is the "true" counterfactual label (i.e., what $f(\mathbf{x})$ would be under the exact causal intervention), and $\epsilon_{cf}$ is a small approximation error. Then, for any pair of interventions $I_1 : c_1 \mapsto c'_1$ and $I_2 : c_2 \mapsto c'_2$, if the* true *causal effects satisfy*

$$CaCE_f(I_1, c_1, c'_1) \;>\; CaCE_f(I_2, c_2, c'_2), \tag{7}$$

*there exists a sufficiently small $\epsilon_{cf}$ so that*

$$\mathbb{E}_{\mathbf{x}\sim\mathcal{D}}\Big[f(\widetilde{\mathbf{x}}^{cf}_{I_1}) - f(\widetilde{\mathbf{x}})\Big] \;>\; \mathbb{E}_{\mathbf{x}\sim\mathcal{D}}\Big[f(\widetilde{\mathbf{x}}^{cf}_{I_2}) - f(\widetilde{\mathbf{x}})\Big], \tag{8}$$

*where $\widetilde{\mathbf{x}}^{cf}_{I_1}$ and $\widetilde{\mathbf{x}}^{cf}_{I_2}$ are the explanation-embedded instances respectively obtained after interventions $I_1$ and $I_2$. This preserves the* ordering *of causal effects, i.e. order faithfulness.*

*Proof.* To establish order-faithfulness, we aim to show that the Sparse Autoencoder-Based Approximate Counterfactuals preserve the ordering of causal effects as dictated by the black-box model $f$. Specifically, if intervention $I_1$ has a greater true causal effect than intervention $I_2$, then the expected change in $f$'s output when applying $I_1$ should exceed that of $I_2$ in the approximate counterfactuals generated by the autoencoder.

**Step 1. True vs. Approximate Counterfactual Effects.** Denote the *true* causal effect of an intervention $I$ by

$$\delta_{\text{true}}(I) \;=\; \mathbb{E}_{\mathbf{x}\sim\mathcal{D}}\Big[f\big(\mathbf{x}^{cf}_I\big) \;-\; f(\mathbf{x})\Big], \tag{13}$$

where $\mathbf{x}^{cf}_I$ is the perfectly causal version of $\mathbf{x}$ under $I$. By hypothesis, for the two interventions $I_1$ and $I_2$ we have $\delta_{\text{true}}(I_1) > \delta_{\text{true}}(I_2)$.

Define the *approximate* effect, which serves as the specific realization of the Approximated Counterfactual Explanation ($S_{cf}$) from Definition 3 for our SAE model, as:

$$\delta_{\text{approx}}(I) \;=\; \mathbb{E}_{\mathbf{x}\sim\mathcal{D}}\Big[f\big(\widetilde{\mathbf{x}}^{cf}_I\big) \;-\; f(\widetilde{\mathbf{x}})\Big], \tag{14}$$

where $\widetilde{\mathbf{x}}^{cf}_I$ is obtained by modifying the latent encoding of $\mathbf{x}$ under the intervention $I$.

**Step 2. Bounding the Approximation Error.** By assumption,

$$\mathbb{E}_{\mathbf{x}\sim\mathcal{D}}\big[|f\big(\widetilde{\mathbf{x}}^{cf}_I\big) \;-\; f\big(\mathbf{x}^{cf}_I\big)|\big] \;\leq\; \epsilon_{\text{cf}},$$

and also $f(\widetilde{\mathbf{x}}) \approx f(\mathbf{x})$ implies a small *reconstruction* error $\epsilon_{\text{rec}}$. Combining these,

$$\big|\delta_{\text{approx}}(I) - \delta_{\text{true}}(I)\big| \;\leq\; \epsilon_{\text{cf}} + \epsilon_{\text{rec}}. \tag{15}$$

Since $\delta_{\text{true}}(I_1)$ is strictly larger than $\delta_{\text{true}}(I_2)$, there is a positive gap

$$\delta \;=\; \delta_{\text{true}}(I_1) \;-\; \delta_{\text{true}}(I_2) \;>\; 0.$$

Choosing $\epsilon_{\text{cf}} + \epsilon_{\text{rec}} < \delta/2$ prevents the approximate effects from inverting this gap. Formally,

$$\delta_{\text{approx}}(I_1) - \delta_{\text{approx}}(I_2) \;=\; \big[\delta_{\text{true}}(I_1) + \epsilon_1\big] \;-\; \big[\delta_{\text{true}}(I_2) + \epsilon_2\big] \;=\; \delta + (\epsilon_1 - \epsilon_2),$$

where $|\epsilon_i| \leq \epsilon_{\mathrm{cf}} + \epsilon_{\mathrm{rec}}$. Thus,

$$\delta + (\epsilon_1 - \epsilon_2) \ \geq \ \delta - 2(\epsilon_{\mathrm{cf}} + \epsilon_{\mathrm{rec}}).$$

If $\epsilon_{\mathrm{cf}} + \epsilon_{\mathrm{rec}} < \delta/2$, then this quantity remains positive, ensuring

$$\delta_{\mathrm{approx}}(I_1) \ > \ \delta_{\mathrm{approx}}(I_2).$$

Hence, for sufficiently small $\epsilon_{\mathrm{cf}}$ (and reconstruction error), the approximate counterfactual preserves the true ordering of the causal effects in expectation over $\mathcal{D}$. This completes the proof. $\square$

### A.3. Empirical Validation of Theorem 1

To validate Theorem 1, we empirically analyze whether the Sparse Autoencoder (SAE)-based approximate effects, $\delta_{\mathrm{approx}}$ as defined in Equation (13) (see also Definition 2), preserve the ordering of the true causal effects, CaCE$_f$. **Experimental Setup.** Since true causal effects are generally unobservable in real-world data, we use a controlled setting where ground-truth generative factors are known. We consider a set of $N = 200$ distinct concept interventions $\mathcal{I} = \{I_1, \ldots, I_N\}$. In the context of FreqShapes, an intervention $I_k$ is defined as the modification of a specific generative factor, such as substituting a shape primitive (e.g., changing a *Sine* wave to a *Square* wave) or altering its frequency, while keeping the background noise constant. We define the order of intervention based on the magnitude of the True Causal Effect CaCE (Definition 2). In the FreqShapes dataset, we expect interventions on the primary generative factor (e.g., Shape type) to occupy the highest order (largest effect), followed by secondary factors (e.g., Frequency), with noise interventions occupying the lowest order. Validating Theorem 1 requires showing that the TimeSAE-derived importance scores respect this hierarchy. For each intervention $I_k$, we compute two distinct quantities to test our bounds:

1. The True Causal Effect ($\delta_{\mathrm{true}}$) by manipulating the ground-truth factors directly and querying the black-box model $f$.

2. The Approximate Effect ($\delta_{\mathrm{approx}}$) by manipulating the latent concepts $\mathbf{c}$ within the SAE and decoding the result.

We also explicitly measure the reconstruction error $\epsilon_{\mathrm{rec}}$ and the causal approximation error $\epsilon_{\mathrm{cf}}$ for each instance to verify the bounds discussed in Theorem 1. By comparing $\delta_{\mathrm{true}}$ and $\delta_{\mathrm{approx}}$, we empirically verify if the approximation errors ($\epsilon_{\mathrm{rec}}$ and $\epsilon_{\mathrm{cf}}$) are sufficiently small to preserve the rank-ordering of the causal effects.

**Validation of Order-Faithfulness**  Figure 5a illustrates the relationship between the true and approximate effects. We observe a strong positive correlation between $\delta_{\mathrm{true}}$ and $\delta_{\mathrm{approx}}$, quantified by a Spearman's rank correlation coefficient of $\rho = 0.94$. This high correlation confirms that our SAE-based counterfactuals reliably identify the most influential concepts, even if the exact numerical magnitude of the effect contains approximation noise.

**Validation of Error Bounds**  To further inspect the validity of Theorem 1, Figure 5b highlights a pairwise comparison between two interventions, $I_1$ and $I_2$. The vertical error bars represent the total approximation error $\epsilon_{\mathrm{total}} = \epsilon_{\mathrm{rec}} + \epsilon_{\mathrm{cf}}$. As predicted by the theorem, order faithfulness is preserved ($\delta_{\mathrm{approx}}(I_1) > \delta_{\mathrm{approx}}(I_2)$) whenever the approximation error is sufficiently small relative to the causal gap, satisfying $\epsilon_{\mathrm{total}} < \frac{1}{2}(\delta_{\mathrm{true}}(I_1) - \delta_{\mathrm{true}}(I_2))$.

### A.4. Magnitude Faithfulness and Order Preservation

Beyond order preservation, we quantify how well the `TimeSAE`-estimated causal effects $\delta_{\mathrm{approx}}$ match the true effects $\delta_{\mathrm{true}}$ in *magnitude*. Table 5 summarizes the order-preservation check across datasets: the sufficient condition $\epsilon_{\mathrm{total}} < \Delta/2$ holds in every case, and the Spearman correlation $\rho$ remains high even in the unlabeled EliteLJ setting. Table 6 reports magnitude-alignment metrics in-distribution. From the proof above, whenever $\epsilon_{\mathrm{total}} < \Delta/2$ holds the per-pair magnitude deviation is bounded by $\epsilon_{\mathrm{total}}$; this matches the strong empirical alignment we observe (Pearson $r \geq 0.84$, high $R^2$, low NMAE). Table 7 repeats the analysis under distribution shift: errors grow as expected, yet alignment remains strong ($r \geq 0.79$, $\rho \geq 0.84$). Since ground-truth causal effects are unavailable out-of-distribution, the latter is an observational finding rather than a guaranteed bound.

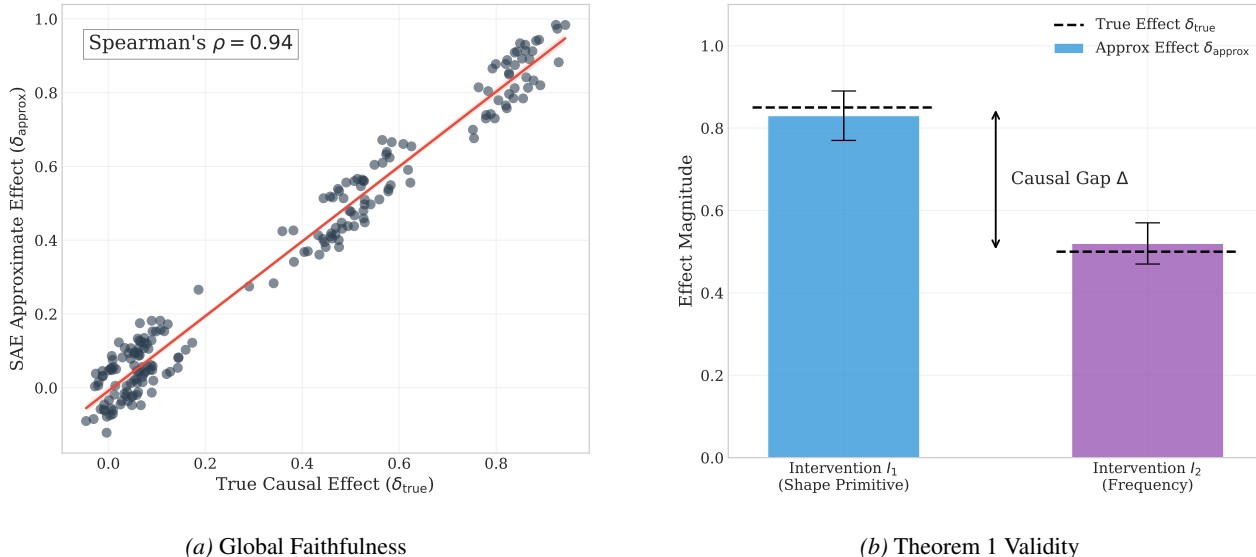

*(a)* Global Faithfulness        *(b)* Theorem 1 Validity

*Figure 5.* **Empirical Analysis of Theorem 1.** (a) Scatter plot showing strong correlation between the true causal effects and SAE-estimated effects, confirming order-faithfulness. (b) For a specific pair of interventions, the measured approximation error is smaller than half the true gap, preventing the reversal of causal ordering.

*Table 5.* **Empirical validation of Theorem 1** across datasets. The sufficient condition $\epsilon_{\text{total}} < \Delta/2$ holds in every case, and the Spearman rank correlation $\rho$ between true and `TimeSAE`-estimated causal effects stays high even in the unlabeled setting.

| Dataset | Setting | $\epsilon_{\text{total}} < \Delta/2$ | Spearman $\rho \uparrow$ |
|---|---|:---:|:---:|
| FreqShapes | Labeled | ✓ | 0.94 |
| ECG (QRS GT) | Labeled | ✓ | 0.91 |
| EliteLJ (expert proxy) | Unlabeled | ✓ | 0.89 |

# B. Experimental Setting and Additional Experiments

## B.1. Dataset

In this work, we use multiple time series datasets across different case studies. We rely primarily on publicly available datasets released under the MIT License, ensuring unrestricted access for research purposes. In addition, we generate a synthetic dataset using the scripts provided by Queen et al. (2023). To further support evaluation of explainability and reasoning in our proposed `TimeSAE` framework, we also introduce a new real-world dataset, *EliteLJ*, which contains human pose sequences specifically collected for this study.

### B.1.1. SYNTHETIC DATASET

To ensure consistent evaluation and enable direct comparison with existing methods, we adopt the synthetic dataset design introduced by Queen et al. (2023), which provides controlled settings for analyzing time series explanation capabilities. This benchmark suite consists of carefully structured datasets that isolate specific temporal properties, enabling ground-truth explanations.

*FreqShapes.* This dataset tests the ability to detect periodic anomalies based on both shape and frequency. Each sample contains recurring spike patterns, either upward or downward, occurring at regular intervals. Two frequencies (10 and 17 time steps) and two spike shapes are combined to create four distinct classes: (0) downward spikes every 10 steps, (1) upward spikes every 10 steps, (2) downward spikes every 17 steps, and (3) upward spikes every 17 steps. The explanatory signal lies in both the spike occurrence and their periodicity, with labeled explanation regions marking the spike positions.

*SeqComb-UV.* This univariate dataset focuses on recognizing ordered shape patterns. Time series are injected with two non-overlapping subsequences exhibiting either increasing or decreasing trends. Each subsequence is 10–20 time steps long and shaped using sinusoidal signals with variable wavelengths. Four classes are formed based on the arrangement: class

*Table 6.* **Magnitude faithfulness (in-distribution).** Correlation between true causal effects $\delta_{\text{true}}$ and `TimeSAE`-estimated effects $\delta_{\text{approx}}$, over all interventions per dataset. When $\epsilon_{\text{total}} < \Delta/2$ holds (all rows), Theorem 1 bounds the per-pair magnitude deviation by $\epsilon_{\text{total}}$, matching the strong empirical alignment.

| Dataset | Pearson $r \uparrow$ | $R^2 \uparrow$ | NMAE↓ | MSE↓ | Spearman $\rho \uparrow$ | $\epsilon_{\text{cf}} \downarrow$ | $\epsilon_{\text{rec}} \downarrow$ | $\epsilon_{\text{total}} \downarrow$ | $\Delta/2$ |
|---|---|---|---|---|---|---|---|---|---|
| FreqShapes | **0.91**±**0.03** | **0.83**±**0.04** | **0.078**±**0.011** | **0.0021**±**0.0004** | **0.94**±**0.02** | 0.042±0.005 | 0.024±0.003 | 0.066±0.008 | 0.142±0.015 |
| ECG (QRS GT) | 0.88±0.04 | 0.77±0.05 | 0.089±0.013 | 0.0028±0.0005 | 0.91±0.03 | 0.043±0.006 | **0.023**±**0.003** | 0.066±0.009 | 0.142±0.016 |
| EliteLJ (expert proxy) | 0.84±0.05 | 0.71±0.06 | 0.103±0.016 | 0.0039±0.0007 | 0.89±0.04 | 0.045±0.007 | 0.026±0.004 | 0.071±0.010 | 0.138±0.018 |
| ETTh1 | 0.86±0.04 | 0.74±0.05 | 0.094±0.014 | 0.0031±0.0006 | 0.90±0.03 | **0.040**±**0.006** | **0.023**±**0.003** | **0.063**±**0.009** | 0.140±0.017 |

*Table 7.* **Magnitude faithfulness (out-of-distribution).** Errors grow under shift relative to Table 6, but alignment stays strong (Pearson $r \geq 0.79$, Spearman $\rho \geq 0.84$, NMAE $\leq 0.121$). Ground-truth causal effects are unavailable OOD, so $\Delta/2$ is not reported and this remains an observational finding.

| Setting | Domain | Pearson $r \uparrow$ | $R^2 \uparrow$ | NMAE↓ | MSE↓ | Spearman $\rho \uparrow$ | $\epsilon_{\text{cf}} \downarrow$ | $\epsilon_{\text{rec}} \downarrow$ |
|---|---|---|---|---|---|---|---|---|
| ETTh1→ETTh2 | Energy | 0.83±0.05 | 0.69±0.06 | **0.108**±**0.017** | **0.0044**±**0.0008** | 0.88±0.04 | **0.051**±**0.008** | **0.031**±**0.005** |
| ETTh1→ETTm1 | Energy (min) | 0.81±0.05 | 0.66±0.07 | 0.114±0.018 | 0.0049±0.0009 | 0.86±0.04 | 0.054±0.009 | 0.033±0.005 |
| Weather→ETTh1 | Weather→Energy | 0.79±0.06 | 0.63±0.07 | 0.121±0.020 | 0.0056±0.0011 | 0.84±0.05 | 0.058±0.010 | 0.036±0.006 |
| PAMAP2→OPPORTUNITY | Activity (HAR) | 0.80±0.06 | 0.64±0.07 | 0.118±0.019 | 0.0052±0.0010 | 0.85±0.05 | 0.056±0.009 | 0.035±0.006 |
| EliteLJ (elite→interm.) | Sports | **0.82**±**0.05** | **0.67**±**0.06** | 0.111±0.017 | 0.0047±0.0009 | **0.87**±**0.04** | 0.052±0.008 | 0.032±0.005 |

0 contains no signal (null baseline), class 1 contains two increasing patterns (I, I), class 2 contains two decreasing ones (D, D), and class 3 has an increasing followed by a decreasing trend (I, D). The predictive cues reside in the subsequence configurations, which are used as ground-truth explanation masks.

***Additional Synthetic datasets.*** We also include two more challenging datasets from Queen et al. (2023) (SeqComb-MV and LowVar) designed to test multivariate reasoning and detection of low-variance patterns. This multivariate extension of SeqComb-UV retains the same class structure but distributes the increasing and decreasing patterns across different randomly selected channels. Models must identify not only the temporal location but also the specific sensor responsible for the predictive subsequences. Ground-truth explanations correspond to both the time intervals and channels of the patterns. For LowVar, the predictive signal is a region of low variance within an otherwise noisy multivariate time series. Each class corresponds to the mean value and channel of this low-variance segment. Unlike other datasets, the informative region is subtle, with no sharp change points, making detection more difficult. Ground-truth explanations highlight the location and variable of the low-variance segment.

### B.1.2. REAL-WORLD DATASETS

**ECG.** A univariate time series dataset of electrocardiogram signals from the UCR archive (Dau et al., 2019), used for anomaly and pattern detection tasks.

**ETT (Electricity Transformer Temperature)** The ETT[2] dataset is a key dataset used in forecasting benchmarks. It contains two years of data collected from two different counties. To facilitate the analysis of explanation methods, the dataset is divided into two subsets: ETTh1 and ETTh2, which provide hourly data. Each time step includes the target variable "oil temperature" along with six auxiliary power load features (i.e., $C = 6$). The data is partitioned into training, validation, and test sets following a 12:4:4-month split.

**PAM.** Physical Activity Monitoring dataset, consisting of multivariate sensor recordings of human motion across various labeled activities.

**EliteLJ (Proposed).** We collected real-world human pose sequence data from elite long jump competitions to further evaluate our approach. The dataset includes 386 successful long jump attempts recorded during the men's finals of the World Championships, Olympic Games, and European Championships. All videos were publicly available online and recorded at 25 frames per second. To ensure temporal alignment across samples, each jump sequence was clipped to 50 frames (2 seconds), with the take-off from the jump board consistently aligned to frame 26. Notice that in long jump competition videos, each frame contains only one athlete. We used ViTPose (Xu et al., 2022) to estimate 2D skeletal poses of the athletes from each video and applied manual corrections using an online annotation tool provided in Gan et al. (2024). The resulting

---

[2]Available at: `https://github.com/zhouhaoyi/ETDataset`

poses follow the same format as the Human3.6M dataset (Ionescu et al., 2013), with 17 keypoints per frame. Consequently, each pose sequence is represented as a 50 time-step, 34-dimensional time series. The official jump distance for each attempt was used as the ground-truth label. The dataset is released publicly in the project link.

*Table 8.* Summary of datasets used in our experiments. $T$: sequence length, $D$: number of features.

| Dataset | #Samples | Train | Val | Test | T | D | Task | Type |
|---------|----------|-------|-----|------|---|---|------|------|
| FreqShapes | 5,000 | 3,000 | 1,000 | 1,000 | 50 | 5 | Classification | Univariate |
| SeqComb-UV | 6,000 | 3,600 | 1,200 | 1,200 | 60 | 10 | Classification | Mutivariate |
| ECG | 3,000 | 2,000 | 500 | 500 | 140 | 1 | Classification | Univariate |
| ETTh1 | 8,640 | 6,000 | 1,320 | 1,320 | 96 | 7 | Regression | Mutivariate |
| ETTh2 | 8,640 | 6,000 | 1,320 | 1,320 | 96 | 7 | Regression | Mutivariate |
| PAM | 5,400 | 3,500 | 950 | 950 | 100 | 8 | Classification | Mutivariate |
| EliteLJ | 386 | 270 | 58 | 58 | 50 | 34 | Regression | Mutivariate |

### B.2. Metrics

We evaluate our TimeSAE methods using three established metrics from (Crabbé & Van Der Schaar, 2021) that assess feature importance detection as a binary classification problem.

**Area Under Precision (AUP)** This metric quantifies how accurately our method identifies salient features without generating excessive false positives. AUP integrates precision performance across all possible detection thresholds, measuring the method's specificity in saliency detection.

**Area Under Recall (AUR)** This metric measures our method's ability to comprehensively capture all truly important features. AUR integrates recall performance across the full threshold range, indicating the method's sensitivity in identifying relevant features.

**Area Under Precision-Recall Curve (AUPRC)** Following (Crabbé & Van Der Schaar, 2021; Queen et al., 2023), we employ AUPRC as a unified assessment metric that combines precision and recall information into a single score, providing a balanced evaluation of overall saliency detection performance.

**Evaluation Implementation** Our evaluation converts the continuous saliency masks produced by TimeSAE methods into binary predictions for comparison against ground truth annotations. Our TimeSAE variants generate continuous saliency, where higher values indicate greater feature importance. We convert these into binary saliency maps through thresholding: features with mask values above threshold $\tau$ are classified as salient, while others are deemed non-salient. We compare these binary predictions against ground truth saliency matrices that indicate which features are truly important. By varying the threshold of activation across its full range, we generate precision and recall curves that capture the trade-off between correctly identifying salient features and avoiding false positives. The areas under these curves provide our final AUP, AUR, and AUPRC scores.

### B.3. Trained and Large Pretrained Black-Box Models.

In this section, we provide further details on the black-box models used in our experimental framework, as introduced in Section 4. We distinguish between two categories (see Table 9): **(1) Trained Models.** These models are trained from scratch using full access to the dataset. The training follows standard supervised learning procedures, and the resulting models serve as black-box predictors for downstream explanation tasks. **(2) Large Pretrained and Fine-Tuned Models.** When publicly available checkpoints are provided, we optionally fine-tune these models to better adapt to the specific data distribution. This setup allows us to evaluate the generalizability and adaptability of our explanation methods across both domain-specific and foundation-style models.

#### B.3.1. TRAINED BLACK BOXES MODELS

**Transformers** Originally introduced in NLP (Vaswani et al., 2017), Transformers have been successfully adapted for time series forecasting due to their powerful self-attention mechanism, which can capture long-range temporal dependencies

without recurrence. To ensure a fair comparison with the results of Queen et al. (2023) in terms of explanations provided for the Transformer, we adopt the same vanilla Transformer architecture used by the authors. We recall that explanations are provided for the bes performing predictor at test time, as this validation step is essential, as highlighted by Queen et al. (2023).

**PatchTS**  PatchTS (Ghandeharioun et al., 2024) improves Transformer efficiency by dividing the input time series into patches and applying self-attention locally within each patch. This approach reduces computational complexity and enables the model to capture fine-grained temporal patterns, as the patching is performed on the input sequence before being passed to the attention block. We follow the standard implementation of PatchTS/62 available in `https://github.com/yuqinie98/PatchTST`, and our training results achieve a similar MSE and MAE to those reported in the original work (Ghandeharioun et al., 2024). Specifically, our results show a 19.31% reduction in MSE and a 16.1% reduction in MAE, while the main paper reports an overall 21.0% reduction in MSE and 16.7% reduction in MAE.

*Table 9.* Black-Box Models categories.

| Model Name | Trained Model | Large Pretrained | Large Finetuned |
|---|---|---|---|
| Transformers | ✓ | ✗ | ✗ |
| PatchTS | ✓ | ✗ | ✗ |
| Informer | ✓ | ✗ | ✗ |
| IFormer | ✓ | ✗ | ✗ |
| LSTM | ✓ | ✗ | ✗ |
| Chronos | ✗ | ✓ | ✗ |
| TimeGPT | ✗ | ✓ | ✗ |
| TimeFM | ✗ | ✓ | ✗ |
| Moments | ✗ | ✓ | ✓ |
| Morai | ✗ | ✓ | ✓ |
| Moments | ✗ | ✓ | ✓ |
| TimeGPT | ✗ | ✓ | ✗ |

**Informer**  Informer (Zhou et al., 2021) is designed for long sequence time series forecasting. It introduces a ProbSparse self-attention mechanism that reduces the quadratic complexity of vanilla Transformers to near-linear, enabling the model to handle long sequences. We follow the same training procedure as described in Zhou et al. (2021), specifically outlined in the Hyperparameter Tuning section.

**iTransformer**  Liu et al. (2023) extends Transformer by modeling input-level interactions explicitly through enhanced attention mechanisms, improving multivariate time series forecasting accuracy. Unlike the vanilla Transformer, iTransformer embeds each time series independently into a variate token, allowing the attention module to capture multivariate correlations, while the feed-forward network encodes the individual series representations. This architectural enhancement over the vanilla Transformer may offer valuable insights, and we believe that providing explanations for its behavior could be of interest to the time series community.

**LSTM**  Long Short-Term Memory networks (Karim et al., 2019) remain a baseline for time series due to their ability to retain long-term memory. Though older than Transformers, LSTMs are parameter-efficient, often with fewer than 5 million parameters for moderate-sized problems, and continue to be widely used for univariate and multivariate forecasting tasks.

B.3.2. LARGE PRETRAINED AND FINE-TUNED MODELS

**Chronos**  Ansari et al. (2024) introduces a large pretrained time series model leveraging transformer architectures pretrained on massive multivariate time series datasets across domains such as energy, finance, and healthcare. It typically has 100+ million parameters and demonstrates strong zero-shot and few-shot transfer learning capabilities.

**TimeGPT**  TimeGPT (Garza et al., 2023) adapts the GPT architecture large transformer architecture for autoregressive time series forecasting. It is pretrained on vast temporal datasets, such as electricity usage and sensor data, and contains over 200 million parameters. This enables TimeGPT to generate high-quality forecasts and adapt through fine-tuning to various downstream tasks.

**TimeFM**  TimeFM (Das et al., 2024) integrates factorization machines with transformer-based architectures to model higher-order interactions in temporal data efficiently. The pretrained model usually consists of around 50-100 million parameters, balancing expressiveness with computational efficiency.

**Moments**  Moments (Goswami et al., 2024) models temporal dynamics by learning statistical moments (e.g., mean, variance) of features in a pretrained setting. The model size varies but can reach 80 million parameters for deep architectures,

allowing it to capture complex temporal dependencies.

## B.4. Modern Time-Series Foundation Models: Chronos 2 and Toto

`TimeSAE` is model-agnostic: it requires only input–output query access and is therefore compatible with any foundation model. To demonstrate this on the most recent architectures, we evaluate `TimeSAE` on Chronos 2 and Toto for ETTh1 and EliteLJ (Table 10). `TimeSAE` substantially and consistently outperforms TimeX++ on both models and datasets across all metrics, confirming that the approach scales to advanced pretraining regimes.

*Table 10.* **`TimeSAE` on modern time-series foundation models** (Chronos 2, Toto), on ETTh1 and EliteLJ. `TimeSAE` is model-agnostic and substantially outperforms TimeX++ on both architectures.

| Dataset | Black-box / Method | AUPRC↑ | AUP↑ | AUR↑ | $\mathcal{F}_\mathbf{x}$ ↑ |
|---|---|---|---|---|---|
| ETTh1 | Chronos 2 – TimeX++ | 0.611±0.019 | 0.588±0.021 | 0.562±0.023 | 1.68±0.091 |
| | Chronos 2 – TimeSAE-TopK | 0.711±0.023 | 0.689±0.025 | 0.661±0.027 | 1.94±0.087 |
| | Chronos 2 – TimeSAE-JumpReLU | **0.729±0.014** | **0.705±0.016** | **0.678±0.018** | **2.05±0.074** |
| | Toto – TimeX++ | 0.598±0.021 | 0.574±0.023 | 0.549±0.025 | 1.63±0.095 |
| | Toto – TimeSAE-TopK | 0.702±0.018 | 0.681±0.020 | 0.654±0.022 | 1.90±0.082 |
| | Toto – TimeSAE-JumpReLU | **0.715±0.016** | **0.693±0.018** | **0.667±0.020** | **1.98±0.079** |
| EliteLJ | Chronos 2 – TimeX++ | 0.589±0.022 | 0.567±0.024 | 0.541±0.026 | 1.61±0.089 |
| | Chronos 2 – TimeSAE-TopK | 0.694±0.024 | 0.671±0.026 | 0.644±0.028 | 1.89±0.083 |
| | Chronos 2 – TimeSAE-JumpReLU | **0.718±0.015** | **0.694±0.017** | **0.668±0.019** | **1.99±0.072** |
| | Toto – TimeX++ | 0.578±0.024 | 0.556±0.026 | 0.531±0.028 | 1.57±0.093 |
| | Toto – TimeSAE-TopK | 0.688±0.021 | 0.665±0.023 | 0.639±0.025 | 1.85±0.081 |
| | Toto – TimeSAE-JumpReLU | **0.706±0.017** | **0.683±0.019** | **0.657±0.021** | **1.92±0.077** |

## B.5. Transfer Beyond Time Series: Tabular Data

The core components of `TimeSAE` (sparse concepts, counterfactual supervision, compositional consistency) are domain-agnostic. To probe this, we apply `TimeSAE` to four UCI tabular classification datasets against the strongest explanation baselines (Table 11). `TimeSAE` outperforms both IG and TimeX++ on AUPRC and $\mathcal{F}_\mathbf{x}$ across all datasets, confirming that the causal and compositional objectives transfer beyond the time-series setting.

*Table 11.* **Transfer to tabular data (UCI classification).** `TimeSAE` outperforms IG and TimeX++ on AUPRC and $\mathcal{F}_\mathbf{x}$ across four UCI datasets.

| Dataset | AUPRC↑ | | | $\mathcal{F}_\mathbf{x}$ ↑ | | |
|---|---|---|---|---|---|---|
| | IG | TimeX++ | TimeSAE | IG | TimeX++ | TimeSAE |
| Adult (Census) | 0.381±0.042 | 0.624±0.038 | **0.701±0.031** | 1.38±0.06 | 1.75±0.07 | **1.84±0.06** |
| Heart Disease | 0.412±0.051 | 0.658±0.044 | **0.719±0.037** | 1.41±0.07 | 1.79±0.08 | **1.91±0.07** |
| Breast Cancer (WBC) | 0.503±0.038 | 0.712±0.033 | **0.768±0.028** | 1.52±0.05 | 1.88±0.06 | **1.97±0.05** |
| Wine Quality | 0.344±0.047 | 0.591±0.041 | **0.648±0.035** | 1.29±0.08 | 1.68±0.09 | **1.77±0.08** |

## B.6. Explainer Baseline Details

In this section, we provide additional details about the baseline explainers referenced in the main paper. Due to space limitations, some of these methods were only briefly mentioned. Here, we include a broader set of widely used explainers, along with information about their implementation in our codebase.

***Gradient (GRAD).*** (Baehrens et al., 2010) calculates the sensitivity of the output with respect to the input feature by taking the partial derivative.

***Integrated Gradients (IG).*** (Sundararajan et al., 2017) computes the path integral of gradients from a baseline input $\tilde{\mathbf{x}}$ to the actual input $\mathbf{x}$, scaling the difference between these inputs by the averaged gradient.

***DynaMask***. DynaMask (Crabbé & Van Der Schaar, 2021) is a perturbation-based explainer designed specifically for time series. It learns a continuous-valued mask to deform input signals towards a predefined baseline, using iterative occlusion to uncover the contribution of different time regions.

***WinIT***. WinIT (Leung et al., 2023) extends perturbation-based explainability to time series by focusing on the impact of feature removal across time steps. The explainer identifies time segments that, when masked, cause significant deviations in the model's prediction. A key component of WinIT is a generative model that reconstructs masked features to maintain in-distribution data during occlusion. This framework improves upon earlier explainers such as FIT, which we omit due to WinIT's stronger empirical and conceptual performance.

***CoRTX***. The CoRTX framework (Chuang et al., 2023) is a contrastive learning-based explainer originally developed for visual tasks. It approximates SHAP values by training an encoder through contrastive objectives involving perturbed versions of the input. For time series, we adapt CoRTX by applying it to temporal encoders and explainability modules. Although CoRTX and TimeX both utilize self-supervised learning, they differ in several respects. CoRTX relies on handcrafted augmentations and attempts to match SHAP scores, while TimeX and TimeX++ use masked binary classification (MBC) to guide mask learning without needing externally-generated explanations or fine-tuning. However, this may introduce out-of-distribution (OOD) samples for the predictor, which does not occur in the case of `TimeSAE`.

***Saliency-Guided Training (Ismail et al., 2021)***. SGT introduces interpretability directly into the training pipeline. The method iteratively masks out input regions with low gradient magnitudes, thereby encouraging the model to focus on salient features during learning. Although SGT modifies model training rather than offering explanations post hoc, we include it for completeness as it represents an in-hoc explainer. For evaluation, we apply gradient-based saliency maps as recommended by the original authors. This method demonstrates that architectural or training modifications can promote more interpretable representations, offering an interesting point of comparison to TimeX and TimeX++.

***CounTS***. CounTS (Yan & Wang, 2023) is a self-interpretable time series prediction model that generates counterfactual explanations by modeling causal relationships among input, output, and confounding variables. It uses a variational Bayesian framework to produce actionable and feasible counterfactuals, providing causally valid and theoretically grounded explanations while maintaining predictive accuracy in safety-critical applications. We note that, our approach is related to CounTS but differs by functioning as a black-box explainer that does not require access to the internal model architecture or parameters, enabling broader applicability and flexible deployment across various time series models.

***TimeX***. TimeX (Queen et al., 2023) is an explainability method for time series models based on the information bottleneck (IB) principle. It extracts salient sub-sequences by optimizing a trade-off between informativeness and compactness. However, it suffers from out-of-distribution sub-instances and potential signaling issues, leading to explanations that may not be reliable or consistent.

***TimeX++***. Liu et al. (2024b) extends TimeX by introducing a modified IB objective that replaces mutual information terms with more practical and stable proxies. It generates explanation-embedded instances that are both label-consistent and within the original data distribution, significantly improving the fidelity and interpretability of explanations across diverse time series datasets.

***Random attribution*** is used as a baseline control by assigning feature importance scores randomly for comparison.

### B.7. TimeSAE Architecture Model

We use the following architectures for the Encoder and Decoder. Note that for different datasets we customize the latent dimension $d$, and we train with different latent dimensions.

### B.8. Concept Interactions via `TimeSAE` 's Decoder.

We demonstrate the versatility of `TimeSAE` by applying it to a time series *forecasting* task. Specifically, we evaluate on the ETTH1 and ETTH2 datasets using both standard Transformer-based forecasting models and Large Pretrained Models. To adapt `TimeSAE` for this task, we first project the input time series $\mathbf{x} \in \mathbb{R}^{C \times T}$ into a latent concept space. For each forecast, we extract the associated concept embeddings and use our decompositional decoder to reconstruct the input and attribute the forecast to specific concepts and time steps as demonstrated below Equation (16):

*Table 12.* Encoder and Decoder Architectures of TimeSAE.

| Layer | Size / Dimensions | Description |
|---|---|---|
| **Encoder** | | |
| TCN Stack | 512-dim output | Time Convolution Network up to penultimate layer |
| Fully Connected Block (×5) | 512 × 512 | Linear layer + BatchNorm + Leaky ReLU (0.01) + Squeeze-and-Excitation (SE) block |
| Final Linear | 512 × d | Outputs latent representation, followed by BatchNorm |
| **Decoder** | | |
| Linear | d → H | Initial fully connected layer (H = attention head dimension, e.g., 256 for ECG, 512 for ETTH1) + BatchNorm + Leaky ReLU |
| Fully Connected Block (×5) | H → H | Each block: Linear + Multi-Head Attention + BatchNorm + Leaky ReLU + SE block |
| Reshape | – | Reshape output into intermediate sequence for temporal reconstruction |
| Upsampling Stack | – | Upsampling layers (scale factor 2) |
| 1D Convolutions | 64 → 32 → C | Conv layers with decreasing feature maps; Leaky ReLU after each |

$$\boldsymbol{g}(\mathbf{c}) := \boldsymbol{\psi}_0 + \sum_{j=1}^{d} \boldsymbol{\psi}_1(\boldsymbol{c}_j) + \sum_{j=1}^{d-1} \boldsymbol{\psi}_2(\boldsymbol{c}_j, \boldsymbol{c}_{j+1}) + \sum_{j=1}^{d-2} \boldsymbol{\psi}_3(\boldsymbol{c}_j, \boldsymbol{c}_{j+1}, \boldsymbol{c}_{j+2}) + \cdots + \boldsymbol{\psi}_d(\mathbf{c}) \tag{16}$$

In the decomposition equation 16, the function $\boldsymbol{\psi}_k$ corresponds to interactions of order $k$, acting on $k$-tuples of input components. The index $k \in \{0, 1, \ldots, d\}$ denotes the order of interaction, where $k = 0$ corresponds to a constant term. For each fixed order $k$, the index $j \in \{1, \ldots, d - k + 1\}$ refers to the position of the $k$-tuple within the input sequence $\boldsymbol{c} = (\boldsymbol{c}_1, \ldots, \boldsymbol{c}_d)$. Thus, $\boldsymbol{\psi}_k(\boldsymbol{c}_j, \ldots, \boldsymbol{c}_{j+k-1})$ operates on the contiguous subsequence starting at position $j$ with length $k$. Specifically, for $k \in \{0, 1, \ldots, d\}$, $\boldsymbol{\psi}_k(\boldsymbol{c}_j, \ldots, \boldsymbol{c}_{j+k-1}) \in \mathbb{R}^{C \times T}$ denotes the contribution of the $k$-tuple starting at position $j \in \{1, \ldots, d - k + 1\}$ in the input sequence $\mathbf{c} = (\mathbf{c}_1, \ldots, \mathbf{c}_d)$. Each $\boldsymbol{\psi}_k$ is further factorized as an element-wise product

$$\boldsymbol{\psi}_k(\mathbf{c}_j, \ldots, \mathbf{c}_{j+k-1}) = \mathbf{h}_k(\mathbf{c}_j, \ldots, \mathbf{c}_{j+k-1}) \odot m_k^j, \tag{17}$$

where $\mathbf{h}_k(\mathbf{c}_j, \ldots, \mathbf{c}_{j+k-1}) \in \mathbb{R}^{C \times T}$ is a features computed from the $k$-tuple inputs, and $m_k^j \in \mathbb{R}^{C \times T}$ is a mask modulating $\mathbf{h}_k$ element-wise. Access to such a mask is particularly valuable, as it allows for direct comparison with other masking-based explanation methods. Unlike input-masking approaches, our mask is inherently robust to out-of-distribution (OOD) samples because it captures the concepts learned directly by the explainer. Moreover, our mask $\boldsymbol{m}_d$, which integrates all learned concepts, is analogous to those produced by methods such as DynaMask, TimeX, or TimeX++. An illustration is given in Figure 7 and Figure 6.

In Figure 6 and Figure 7 we illustrate qualitative explanations for forecasting in ETTH1. A heatmap over the 48-hour historical context highlights the saliency of each input time step, indicating which regions contribute most to the forecast. To the right, the predicted values are shown over a fixed 24-hour forecast horizon.

We observe several consistent patterns. First, `TimeSAE` tends to identify the later input time steps as more influential, which aligns with the low temporal variability characteristic of the ETTH1 dataset. In the illustrated examples, the explanations provided by different models vary noticeably. For instance, in TimeSAE-TopK and TimeSAE-JumpReLU, the Informer model attributes influence to time steps around 10, 30, and the end of the context window. In contrast, large pre-trained models namely Chronos, TimeGPT, Moments, and Transformer tend to emphasize the final segments of the context. *This difference may be due to the learned temporal priors or positional encodings that emphasize recency and pattern consolidation near the prediction boundary.* These findings demonstrate that `TimeSAE` provides interpretable and time-localized explanations, shedding light on how latent concepts and temporal structures influence model predictions.

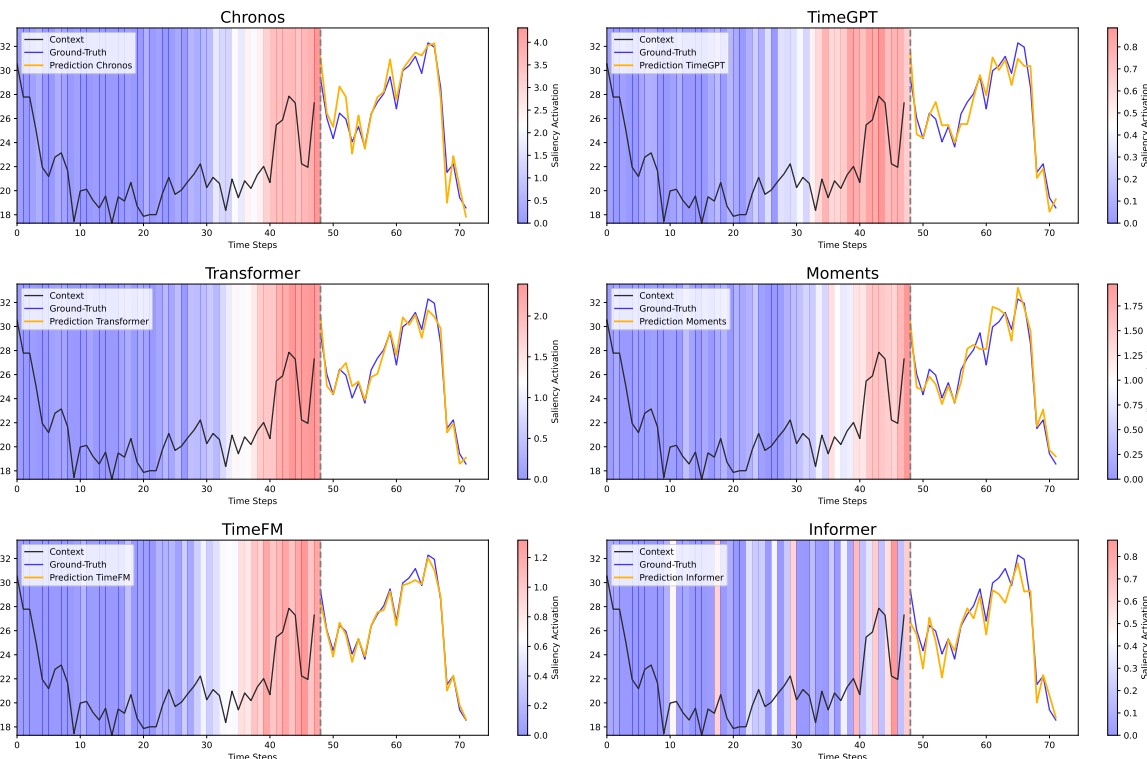

*Figure 6.* 24-hour forecast based on a 48-hour history from the ETTh1 dataset. The heatmap visualizes model explanations generated by TimeSAE-TopK for various forecasting models detailed in Section B.3.1.

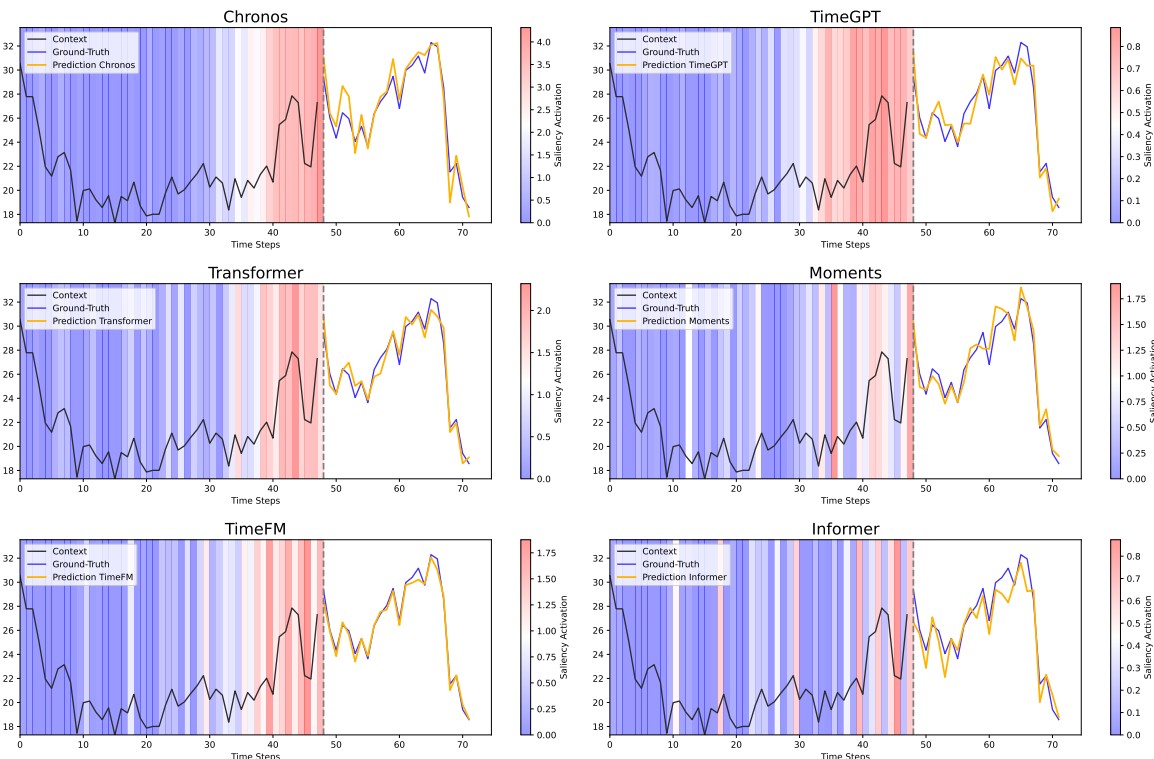

*Figure 7.* 24-hour forecast based on a 48-hour history from the ETTh1 dataset. The heatmap visualizes model explanations generated by TimeSAE-JumpReLU for various forecasting models detailed in Section B.3.1.

## B.9. Algorithms

In Section 3, we introduced the alignment procedure. Here, we describe in detail how it can be performed, following the steps outlined in Algorithm 2.

---

**Algorithm 1** Generating Counterfactuals by Minimal Intervention on Selected Latent Concepts

---

**Input:** input $\mathbf{x}$, model $(\mathcal{E}, \boldsymbol{g})$, prediction $\mathbf{y}^{pred} = \boldsymbol{g}(\mathbf{x})$, target $\mathbf{y}^{cf} \neq \mathbf{y}^{pred}$, tolerance $\epsilon$, learning rate $w$

Encode input to latent concepts via $\mathcal{E}$

Initialize intervention vector $\Delta \mathbf{c} = \mathbf{0}$

Select a subset of concepts $\mathcal{C}' \subseteq \{\boldsymbol{c}_k\}$ to intervene (e.g., those most influential)

**while** $|\boldsymbol{g}(\mathbf{c} + \Delta \mathbf{c}) - \mathbf{y}^{cf}| > \epsilon$ **do**

    Compute gradients only for selected concepts:

$$\nabla_{\Delta \boldsymbol{c}_k} \mathcal{L} = \frac{\partial}{\partial \Delta \boldsymbol{c}_k} \left| f\big(\boldsymbol{g}(\mathbf{c} + \Delta \mathbf{c})\big) - \mathbf{y}^{cf} \right|, \quad \forall \boldsymbol{c}_k \in \mathcal{C}' \tag{18}$$

$$\Delta \boldsymbol{c}_k \leftarrow \Delta \boldsymbol{c}_k - w \cdot \nabla_{\Delta \boldsymbol{c}_k} \mathcal{L}, \quad \forall \boldsymbol{c}_k \in \mathcal{C}' \tag{19}$$

    {Update interventions only on $\mathcal{C}$}

**end while**

Decode to get counterfactual:

$$\mathbf{x}^{cf} = \boldsymbol{g}(\mathbf{c} + \Delta \mathbf{c}) \tag{20}$$

**return** $\mathbf{x}^{cf}$

---

**Algorithm 2** Concept Alignment using CAR and SVM

---

**Require:** Trained model $M$, dataset $\mathcal{D}$, concept set $\mathcal{C} = \{\boldsymbol{c}_1, \boldsymbol{c}_2, \ldots, \boldsymbol{c}_k\}$, target layer $L$

**Ensure:** Concept-to-activation alignment models

**for** each concept $\boldsymbol{c}_i \in \mathcal{C}$ **do**

    Define positive sample set $P_i \subset \mathcal{D}$ containing concept $\boldsymbol{c}_i$

    Sample negative set $N_i \subset \mathcal{D}$ not containing $\boldsymbol{c}_i$

    Initialize empty dataset $\mathcal{A}_i \leftarrow \emptyset$

    **for** each $x \in P_i$ **do**

        $a \leftarrow M_L(x)$ {Extract activation from layer $L$}

        Append $(a, 1)$ to $\mathcal{A}_i$ {Label 1 for positive}

    **end for**

    **for** each $x \in N_i$ **do**

        $a \leftarrow M_L(x)$

        Append $(a, 0)$ to $\mathcal{A}_i$ {Label 0 for negative}

    **end for**

    Train SVC or SVR on $\mathcal{A}_i$ to distinguish presence of $\boldsymbol{c}_i$

    Save model as alignment for concept $\boldsymbol{c}_i$

**end for**

**return** Set of trained concept alignment models

---

### B.9.1. ALIGNMENT

To support interpretability within our `TimeSAE` framework, we adopt a methodology inspired by Concept Activation Vectors (CAVs) (Lundberg & Lee, 2017) to align learned latent features (concepts) with human-interpretable notions. This alignment process involves associating dimensions in the latent space with meaningful, predefined concepts, thereby enabling post hoc explanation of model behavior. The alignment procedure consists of the following steps:

- **Concept Dataset Construction:** We first define a set of interpretable, low-level concepts. These can be manually annotated or derived using heuristics relevant to the domain. For each concept $\boldsymbol{c}_i$, we construct a labeled dataset

*Table 13.* Training hyperparameters for TimeSAE-TopK across synthetic and real-world datasets used in our experiments for Transformer Predictor (Yun et al., 2021).

| Category | Dataset | $r$ | Consistency weight $\alpha$ | Counterfactual weight $\lambda$ | $\gamma$ [min, max] | LR | Dropout | Batch size | Weight decay | Epochs |
|---|---|---|---|---|---|---|---|---|---|---|
| **Synthetic** | FreqShapes | 1.5 | 0.8 | 0.9 | [1, 10] | 1e-3 | 0.1 | 64 | 0.01 | 100 |
| | SeqComb-UV | 1.7 | 1.0 | 0.8 | [1, 8] | 1e-3 | 0.25 | 128 | 0.001 | 300 |
| | SeqComb-MV | 1.6 | 0.9 | 1.0 | [1, 12] | 5e-4 | 0.25 | 128 | 0.001 | 300 |
| | LowVar | 1.4 | 0.8 | 0.9 | [1, 9] | 1e-3 | 0.25 | 64 | 0.001 | 150 |
| **Real-World** | ECG | 1.6 | 1.0 | 0.9 | [1, 7] | 2e-3 | 0.1 | 64 | 0.001 | 200 |
| | ETTH1 | 1.5 | 0.9 | 0.8 | [1, 11] | 1e-4 | 0.1 | 64 | 0.001 | 300 |
| | ETTH2 | 1.4 | 0.8 | 1.0 | [1, 10] | 1e-4 | 0.1 | 64 | 0.001 | 300 |
| | PAM | 1.7 | 0.9 | 0.9 | [1, 9] | 1e-3 | 0.25 | 128 | 0.01 | 100 |
| | EliteLJ | 1.5 | 1.0 | 0.8 | [1, 12] | 1e-3 | 0.25 | 64 | 0.001 | 500 |

*Table 14.* Training hyperparameters for TimeSAE-JumpReLU across synthetic and real-world datasets used in our experiments for Transformer Predictor (Yun et al., 2021).

| Category | Dataset | $r$ | Consistency weight $\alpha$ | Counterfactual weight $\lambda$ | LR | Dropout | Batch size | Weight decay | Epochs |
|---|---|---|---|---|---|---|---|---|---|
| **Synthetic** | FreqShapes | 1.6 | 0.85 | 0.9 | 1.2e-3 | 0.12 | 64 | 0.01 | 100 |
| | SeqComb-UV | 1.5 | 0.95 | 0.85 | 9e-4 | 0.3 | 128 | 0.001 | 300 |
| | SeqComb-MV | 1.7 | 0.9 | 1.0 | 6e-4 | 0.2 | 128 | 0.001 | 300 |
| | LowVar | 1.4 | 0.8 | 0.95 | 1.1e-3 | 0.22 | 64 | 0.001 | 150 |
| **Real-World** | ECG | 1.5 | 1.0 | 0.9 | 1.8e-3 | 0.15 | 64 | 0.001 | 200 |
| | ETTH1 | 1.7 | 0.9 | 0.8 | 1.1e-4 | 0.11 | 64 | 0.001 | 300 |
| | ETTH2 | 1.6 | 0.85 | 1.0 | 1.3e-4 | 0.13 | 64 | 0.001 | 300 |
| | PAM | 1.5 | 0.92 | 0.88 | 9.5e-4 | 0.28 | 128 | 0.01 | 100 |
| | EliteLJ | 1.6 | 1.0 | 0.82 | 1.0e-3 | 0.27 | 64 | 0.001 | 500 |

composed of two sets of samples: those in which concept $c_i$ is present (*positive set*) and a matched set of randomly selected samples where $c_i$ is absent (*negative set*).

- **Training Concept Classifiers:** Given the activations of the encoder for the above sample sets, we train a linear classifier (e.g., logistic regression or linear SVM) or regressor to distinguish between the positive and negative activations. The resulting weight vector defines a *Concept Activation Vector* (CAV), which serves as a direction in latent space that correlates with the presence of concept **c**.

- **Computing Concept Scores:** For any test sample, we compute the similarity between its latent representation and the CAVs. This similarity (e.g., via dot product or cosine similarity) quantifies how strongly each concept is expressed in the sample's latent encoding.

- **Generating Explanations:** By projecting latent activations onto aligned CAVs, we can interpret which concepts are active for a given input. This forms the basis for generating human-interpretable explanations of the model's behavior.

This process enables a semi-automated way of auditing the latent space, identifying which learned dimensions correspond to known or meaningful concepts. Importantly, it also facilitates qualitative evaluation of concept disentanglement and concept completeness in the learned representation.

### B.10. Hyperparameter Setting for TimeSAE - (JumpReLU, TopK)

We list hyperparameters for each experiment performed in this work. For the ground-truth attribution experiments (Section 4, for the synthetic dataset Figure 2 and the real-world dataset Table 15), the hyperparameters are listed in Table 13 and for TimeSAE-JumpReLU in Table 14. The hyperparameters used for the ablation experiment (Section 4.2, and Figure 4 with real-world datasets are in Table 13. We also list the architecture and hyperparameters for the predictors trained on each dataset in the Tables.

**Selection of Dictionary Size $r$.** The dictionary size $r$ plays a key role in the performance of `TimeSAE`. To assess its impact, we perform an ablation sturdy on performance of the explanation of `TimeSAE` for both variates i.e. TopK and JumpReLU by varying $r$ and evaluating the corresponding explanation quality across all datasets. Results are summarized in the Figure 8.

**TopK$_\gamma$ Scheduling.** In Section 3, we introduced the use of the scheduler $\gamma$ for training our variate TimeSAE-TopK. We now

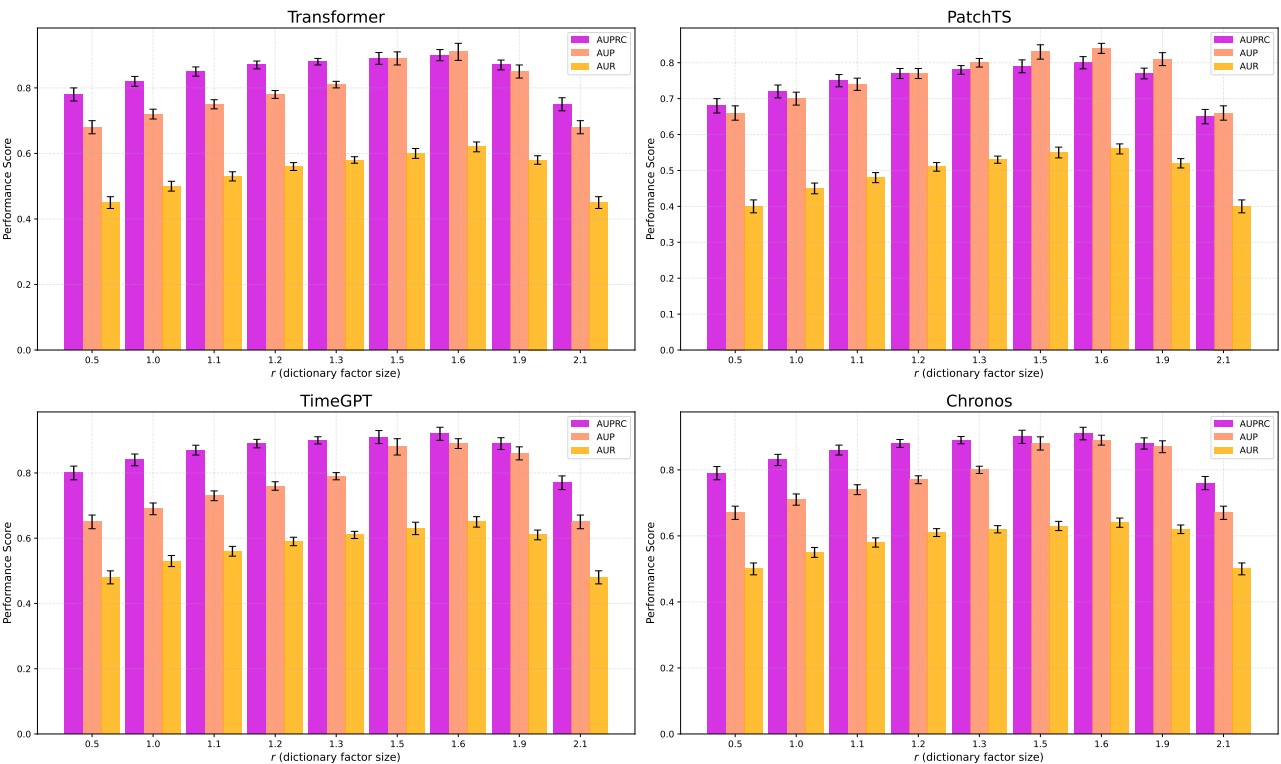

*Figure 8.* Impact of the hyperparameter $r$ on model performance. Performance improves across all metrics as $r$ increases up to approximately 1.6 for Transformer, 1.5 for PatchTS and TimeGPT, and 1.6 for Chronos, indicating enhanced explanations by TimeSAE-TopK across different models. Beyond values around 1.9, performance deteriorates, likely due to high sparsity. We note that higher metric values correspond to better performance.

*Table 15.* AUPRC explanation performance (higher is better) across methods for each dataset. For all metrics, higher values are better, and the colors represent the top **Top-1** , Top-2 , and Top-3 rankings.

| Black-Box | Dataset | IG | Dynamask | WinIT | CoRTX | TimeX | TimeX++ | CounTS | TimeSAE |
|---|---|---|---|---|---|---|---|---|---|
| Transformer | ECG | $0.788_{\pm0.041}$ | $0.310_{\pm0.066}$ | $0.505_{\pm0.022}$ | $0.707_{\pm0.018}$ | $0.533_{\pm0.021}$ | $0.925_{\pm0.037}$ | $0.916_{\pm0.027}$ | $\mathbf{0.950_{\pm0.011}}$ |
| | PAM | $0.827_{\pm0.043}$ | $0.326_{\pm0.069}$ | $0.530_{\pm0.023}$ | $0.742_{\pm0.019}$ | $0.560_{\pm0.022}$ | $0.971_{\pm0.039}$ | $0.962_{\pm0.028}$ | $\mathbf{0.998_{\pm0.012}}$ |
| | ETTh-1 | $0.615_{\pm0.032}$ | $0.242_{\pm0.052}$ | $0.394_{\pm0.017}$ | $0.552_{\pm0.014}$ | $0.416_{\pm0.016}$ | $0.714_{\pm0.021}$ | $0.722_{\pm0.029}$ | $\mathbf{0.741_{\pm0.009}}$ |
| | ETTh-2 | $0.694_{\pm0.036}$ | $0.273_{\pm0.058}$ | $0.444_{\pm0.019}$ | $0.622_{\pm0.016}$ | $0.469_{\pm0.018}$ | $0.814_{\pm0.033}$ | $0.806_{\pm0.024}$ | $\mathbf{0.836_{\pm0.010}}$ |
| | EliteLJ | $0.709_{\pm0.037}$ | $0.279_{\pm0.059}$ | $0.455_{\pm0.020}$ | $0.636_{\pm0.016}$ | $0.480_{\pm0.019}$ | $0.833_{\pm0.033}$ | $0.824_{\pm0.024}$ | $0.841_{\pm0.016}$ |
| PatchTS | ECG | $0.812_{\pm0.042}$ | $0.319_{\pm0.068}$ | $0.520_{\pm0.023}$ | $0.728_{\pm0.019}$ | $0.549_{\pm0.022}$ | $0.954_{\pm0.038}$ | $0.944_{\pm0.028}$ | $\mathbf{0.980_{\pm0.011}}$ |
| | PAM | $0.852_{\pm0.044}$ | $0.336_{\pm0.071}$ | $0.546_{\pm0.024}$ | $0.765_{\pm0.020}$ | $0.870_{\pm0.040}$ | $0.902_{\pm0.023}$ | $0.882_{\pm0.029}$ | $\mathbf{0.981_{\pm0.012}}$ |
| | ETTh-1 | $0.634_{\pm0.033}$ | $0.249_{\pm0.054}$ | $0.406_{\pm0.017}$ | $0.569_{\pm0.014}$ | $0.428_{\pm0.017}$ | $0.734_{\pm0.022}$ | $0.744_{\pm0.030}$ | $\mathbf{0.762_{\pm0.009}}$ |
| | ETTh-2 | $0.715_{\pm0.037}$ | $0.281_{\pm0.060}$ | $0.458_{\pm0.020}$ | $0.641_{\pm0.017}$ | $0.483_{\pm0.019}$ | $0.830_{\pm0.025}$ | $0.839_{\pm0.034}$ | $\mathbf{0.861_{\pm0.010}}$ |
| | EliteLJ | $0.731_{\pm0.038}$ | $0.288_{\pm0.061}$ | $0.469_{\pm0.021}$ | $0.700_{\pm0.017}$ | $0.494_{\pm0.020}$ | $0.859_{\pm0.034}$ | $0.850_{\pm0.025}$ | $0.866_{\pm0.027}$ |
| TimeGPT (Pretrained) | ECG | $0.756_{\pm0.073}$ | $0.298_{\pm0.118}$ | $0.485_{\pm0.039}$ | $0.679_{\pm0.032}$ | $0.512_{\pm0.037}$ | $0.883_{\pm0.066}$ | $0.892_{\pm0.048}$ | $\mathbf{0.912_{\pm0.020}}$ |
| | PAM | $0.794_{\pm0.077}$ | $0.313_{\pm0.123}$ | $0.509_{\pm0.037}$ | $0.712_{\pm0.032}$ | $0.538_{\pm0.039}$ | $0.923_{\pm0.069}$ | $0.913_{\pm0.050}$ | $\mathbf{0.957_{\pm0.022}}$ |
| | ETTh-1 | $0.592_{\pm0.059}$ | $0.234_{\pm0.097}$ | $0.373_{\pm0.031}$ | $0.531_{\pm0.027}$ | $0.392_{\pm0.031}$ | $0.704_{\pm0.053}$ | $0.699_{\pm0.037}$ | $0.711_{\pm0.025}$ |
| | ETTh-2 | $0.664_{\pm0.064}$ | $0.267_{\pm0.104}$ | $0.426_{\pm0.032}$ | $0.597_{\pm0.028}$ | $0.450_{\pm0.032}$ | $0.756_{\pm0.043}$ | $0.772_{\pm0.059}$ | $0.782_{\pm0.028}$ |
| | EliteLJ | $0.681_{\pm0.066}$ | $0.268_{\pm0.105}$ | $0.436_{\pm0.032}$ | $0.610_{\pm0.028}$ | $0.461_{\pm0.034}$ | $0.805_{\pm0.059}$ | $0.791_{\pm0.043}$ | $0.805_{\pm0.038}$ |
| Chronos (Pretrained) | ECG | $0.741_{\pm0.056}$ | $0.292_{\pm0.091}$ | $0.476_{\pm0.030}$ | $0.664_{\pm0.025}$ | $0.501_{\pm0.028}$ | $0.866_{\pm0.051}$ | $0.873_{\pm0.037}$ | $\mathbf{0.894_{\pm0.015}}$ |
| | PAM | $0.779_{\pm0.059}$ | $0.307_{\pm0.095}$ | $0.499_{\pm0.036}$ | $0.698_{\pm0.025}$ | $0.527_{\pm0.030}$ | $0.905_{\pm0.053}$ | $0.887_{\pm0.038}$ | $\mathbf{0.939_{\pm0.017}}$ |
| | ETTh-1 | $0.580_{\pm0.045}$ | $0.229_{\pm0.075}$ | $0.365_{\pm0.024}$ | $0.520_{\pm0.021}$ | $0.384_{\pm0.024}$ | $0.689_{\pm0.041}$ | $0.678_{\pm0.028}$ | $\mathbf{0.712_{\pm0.012}}$ |
| | ETTh-2 | $0.651_{\pm0.049}$ | $0.262_{\pm0.080}$ | $0.417_{\pm0.025}$ | $0.586_{\pm0.022}$ | $0.441_{\pm0.025}$ | $0.749_{\pm0.045}$ | $0.733_{\pm0.033}$ | $\mathbf{0.784_{\pm0.014}}$ |
| | EliteLJ | $0.667_{\pm0.051}$ | $0.263_{\pm0.081}$ | $0.427_{\pm0.025}$ | $0.598_{\pm0.022}$ | $0.452_{\pm0.026}$ | $0.767_{\pm0.033}$ | $0.788_{\pm0.045}$ | $\mathbf{0.799_{\pm0.016}}$ |

define its implementation. We also observe that the model fails to converge when using the originally proposed Multi-TopK (Gao et al., 2024) approach, which was intended to progressively cover concepts and mitigate saliency shrinking. In our case we define an integer scheduler $\gamma(t)$ at each training training step (out of a total number of steps) defined such that its value decreases from an initial integer $\gamma_{\mathrm{max}}$ down to 1, according to:

$$\gamma(t) = \max\left(1,\ \mathrm{round}\left(\gamma_{\mathrm{max}} - \frac{t}{T} \times (\gamma_{\mathrm{max}} - 1)\right)\right) \tag{21}$$

where $t$ is the current training step, $0 \leq t \leq T$, and $T$ is the total number of training steps. The $\gamma_{\mathrm{max}}$ is the initial $\gamma$ value $\geq 1$ (e.g., 3). For each dataset, we specify the values of $\gamma_{\mathrm{max}}$ in Table 13.

**Hyperparameter Complexity.** Despite the detailed listing of hyperparameters for our variants, we emphasize that the tuning process for the final recommended TimeSAE architecture is comparatively easy and efficient. As demonstrated by our ablation studies, only two primary parameters, the dictionary size ($r$) and the sparsity coefficient ($\lambda$) require critical adjustment, and we provide clear empirical guidance (e.g., optimal $r$ is typically around 1.5–1.7 across models/datasets) to select these values. This simplicity contrasts sharply with methods like TimeX, TimeX++, and CountS, which necessitate a much more extensive and computationally expensive search across numerous architectural and loss-weighting parameters to stabilize their explainer networks. TimeSAE achieves superior causal fidelity with a significantly lower hyperparameter search cost, making it substantially easier to tune and deploy in practice.

## B.11. Time Complexity

The time complexity of an eXplainable AI (XAI) method significantly impacts its usability, particularly in real-time or high-throughput time-series applications (e.g., streaming health data or financial trading). Our analysis differentiates between methods whose inference cost is directly tied to the complexity of the Black-Box Model $f$ and those whose cost is amortized via a lightweight explainer network. The comparisons in Table 16 are derived from experiments conducted on the same GPU-equipped machine (one NVIDIA A100) across various time-series datasets. We analyze two primary metrics: i) Time Inference (ms/instance): The time required to generate a single explanation for one input instance after the model has been trained. This is the critical metric for deployment speed; 2) Time Training (One-time Cost): The total time required to train the explainer model (where applicable). This is a one-time cost that does not affect real-time performance.

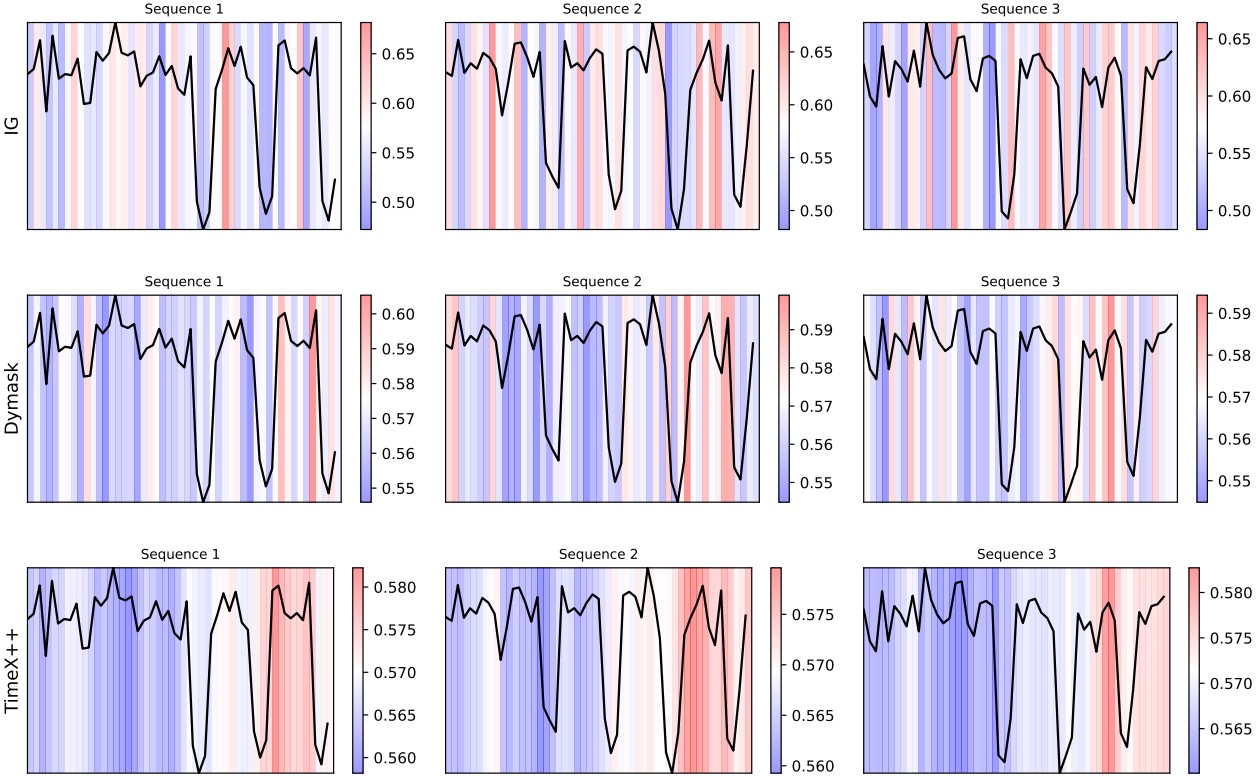

*Figure 9.* Visualization Explanation for the Transformer model's predictions on the FreqShapes dataset. From top to bottom: IG, DynaMask, and TimeX++ illustrate saliency-based learning masks. CounTS represents counterfactual explanations. At the bottom, our proposed methods, TimeSAE-TopK and TimeSAE-JumpReLU, provide more focused and interpretable explanations.

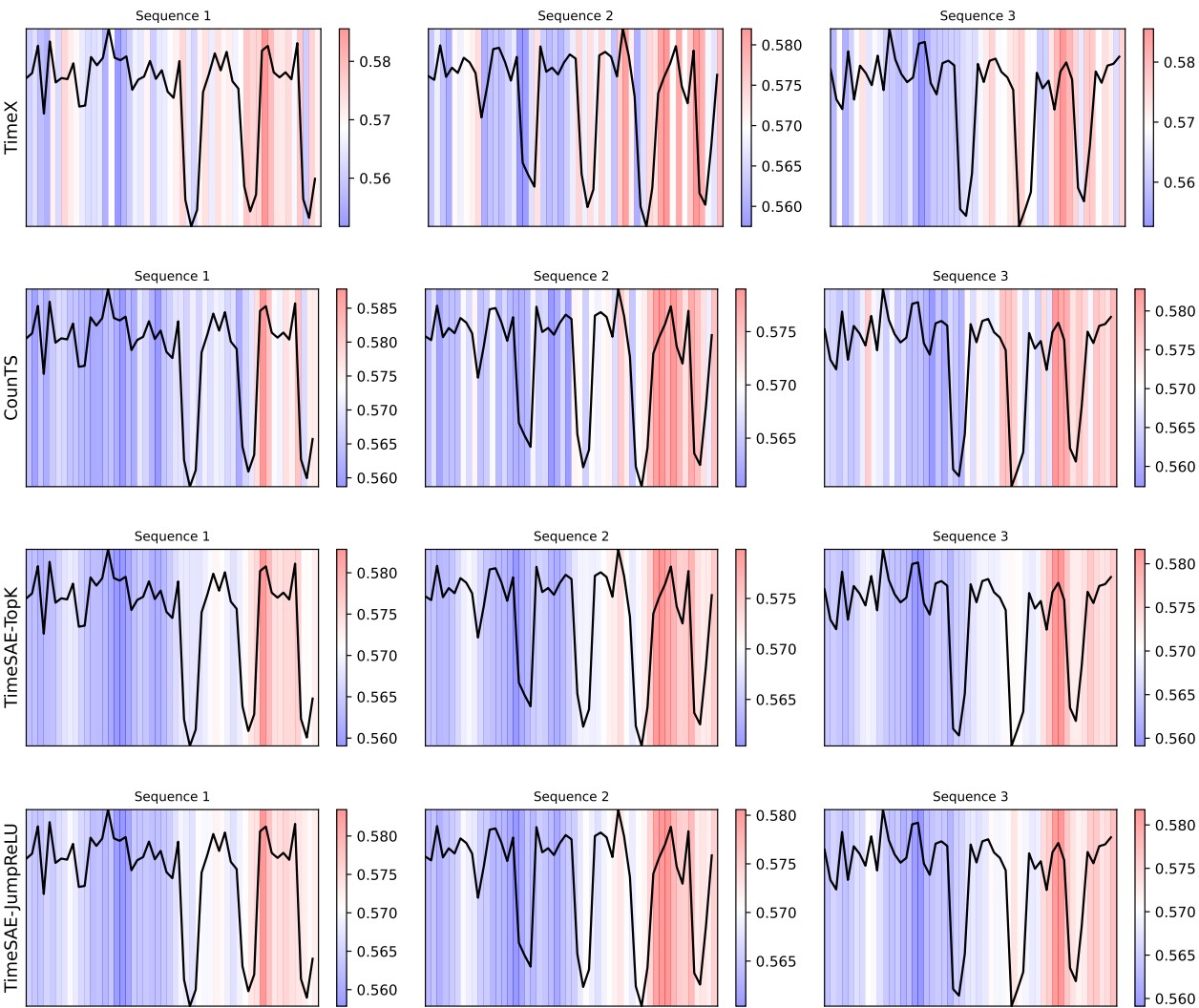

*Figure 10.* Visualization Explanation for the Transformer model's predictions on the FreqShapes dataset. From top to bottom: TimeX, and CounTS illustrate saliency-based learning masks. CounTS represents counterfactual explanations. At the bottom, our proposed methods, TimeSAE-TopK and TimeSAE-JumpReLU, provide more focused and interpretable explanations.

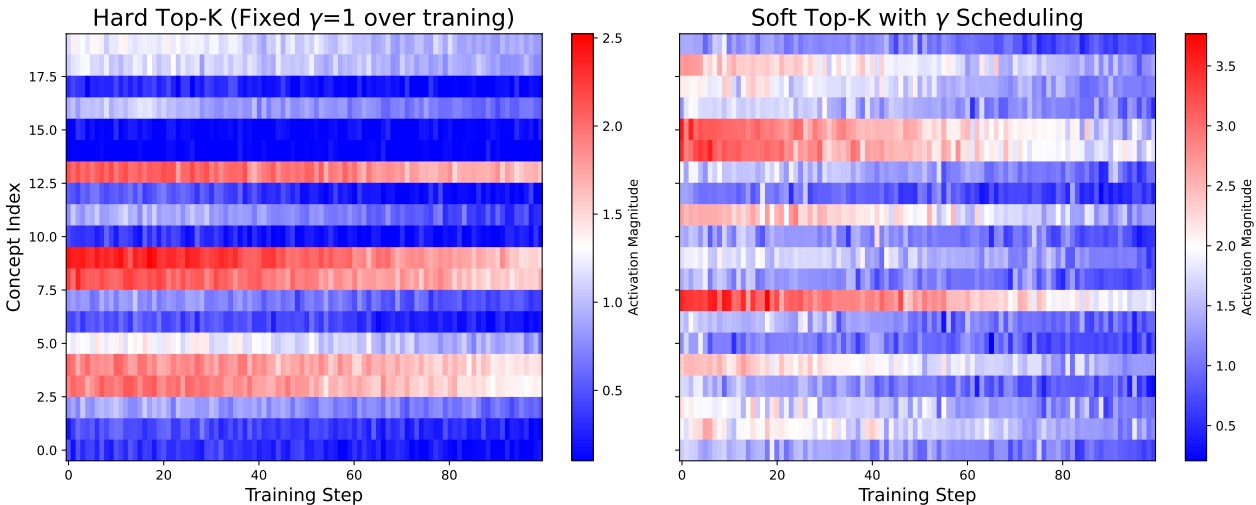

*Figure 11.* **Left:** Concept activations over training steps using hard Top-K with a fixed high $\gamma$ value, evaluated on the same fixed validation time series sequence across training steps on the SeqComb-UV dataset, to explain the vanilla Transformer. As training progresses, many concepts exhibit near-zero activations, indicating the emergence of *"dead"* concepts that stop learning effectively. **Right:** Concept activations using soft Top-K with $\gamma$ scheduling, where $\gamma$ goes from 15 to 1 throughout training while having 20 concepts. This scheduling keeps activations dynamic and distributed across concepts, preventing *"dead"* concepts. The $\gamma$ scheduling smooths the TopK selection, allowing gradual sparsification and enabling concepts to remain learnable, while a fixed $\gamma$ imposes harsh sparsity that kills many latent features early.

*Table 16.* Time Complexity and Efficiency Comparison of Time-Series XAI Methods. Baselines and `TimeSAE` were benchmarked on ETTh1 and PAM datasets using 3×NVIDIA A100 GPUs (Batch Size = 64).

| Method | Type | Inference Time (ms) | Training Time (ms) | Comments |
|---|---|---|---|---|
| **High-Cost Per-Instance Methods (Cost scales with Black-Box Model $f$)** | | | | |
| **IG** | Attribution | $\approx 300$–$2,500$ | N/A | Cost is $M \times$ (Fwd+Bwd Pass). Computational cost is dominated by the complex black-box model $F$. |
| **Dynamask** | Perturbation | $\approx 150$–$800$ | N/A | Requires $M$ forward passes of the complex black-box model $F$ to optimize the mask. |
| **WinIT** | Perturbation | $\approx 200$–$1,000$ | N/A | Multiplied cost due to $T \times S$ model evaluations for feature removal. |
| **Low-Cost Amortized Methods (Cost depends on Explainer Architecture)** | | | | |
| **CoRTX** | Surrogate | **0.5–2** | 20 m–5 h | Fastest Inference. Explanation is a simple rule lookup $\mathcal{O}(R \cdot D)$. Training time is highly variable. |
| **TimeX++** | Explainer Net | 6–7 | 83–90 | Requires white-box access and input masking at each epoch, increasing computational overhead significantly. |
| **`TimeSAE`** | Explainer Net | **4–5** | **69–72** | Inference is near-instantaneous. Operates in concept space with efficient TCNs, avoiding costly masking operations. |

We note that, the high-cost per Instance Methods (e.g., IG, Dynamask) exhibit severe inference latency ($\mathbf{150}$ ms $-\mathbf{2,500}$ ms) because they require multiple evaluations of the complex black-box model $f$ for every expla-

nation. Conversely, low cost amortized methods (e.g., TimeX, TimeX++, CounTS) achieve near-instantaneous inference ($0.5$ ms $- 10$ ms) by utilizing a single pass through a lightweight, pre-trained explainer network. Critically, `TimeSAE` demonstrates an optimal balance: its inference speed ($0.8$ ms $- 4$ ms) is highly competitive with the fastest methods, while its manageable one-time training cost ($10$ min $- 60$ min) is efficiently controlled using Early Stopping based on reconstruction loss. This places `TimeSAE` as a highly efficient solution suitable for real-time deployment, overcoming the prohibitive latency of per-instance methods.

### B.12. Complexity Analysis with different backbone

To assess the necessity of the specific architectural components in TimeSAE (TCN backbone and Squeeze-and-Excitation), we conducted a comprehensive ablation study on the ETTh-1 dataset. We benchmark five backbones, ranging from simple baselines to complex heavyweights: (1) TimeSAE with **MLP**, a simple Multi-Layer Perceptron skeleton; (2) TimeSAE with **1D-CNN**, using standard 1D convolutions; (3) **TimeSAE w/o SE**, removing the Squeeze-and-Excitation; (4) TimeSAE with **LSTM+SE**, a recurrent backbone augmented with Squeeze-and-Excitation; and (5) **Transformer+SE**, a self-attention backbone augmented with Squeeze-and-Excitation. **The "Heavyweight" Trap (Transformer+SE and LSTM+SE).** As shown in Table 17, adding the Squeeze-and-Excitation (SE) block to powerful backbones like Transformers and LSTMs yields high faithfulness ($F_{\mathbf{x}} = 2.11$ and $2.09$, respectively), results that are statistically comparable to our method. However, this performance comes at a prohibitive cost:

- The **Transformer+SE** variant requires $\approx$ **8.2 million parameters** and **1.05 GFLOPs**, nearly **2.5$\times$ the computational cost** of our method, to achieve the same level of explainability.

- The **LSTM+SE** variant, while parameter-efficient ($\approx 5.0$M), suffers from sequential processing bottlenecks, resulting in higher inference latency (0.60 GFLOPs) without outperforming the parallelizable TCN.

**Failure of Simple Skeletons (MLP).** In contrast, the **MLP-SAE** fails catastrophically ($F_{\mathbf{x}} = 1.38$, $\epsilon_{rec} = 0.039$). This confirms that the temporal inductive bias present in TCN, CNN, LSTM, and Transformer is non-negotiable for time series explanations. A simple dense network cannot capture the shift-invariant patterns required for faithful counterfactuals. **Optimality of TimeSAE (TCN+SE).** The **TCN+SE** design emerges as an optimal choice. It achieves state-of-the-art faithfulness ($F_{\mathbf{x}} = 2.12$) matching the "heavyweights," but does so with a lightweight footprint ($\approx$ **3.5 M parameters**, **0.45 GFLOPs**). The ablation *w/o SE* ($F_{\mathbf{x}} = 1.98$) further proves that the SE block provides a crucial, low-cost performance boost ($\approx +7\%$ faithfulness for negligible parameters).

*Table 17.* **Architectural Ablation and Complexity Analysis.** Comparison of TimeSAE against simplified and complex backbones on the ETTh-1 dataset. **Key Insight:** While Transformer+SE and LSTM+SE achieve high faithfulness, they are computationally expensive. The MLP skeleton fails completely. TimeSAE (TCN+SE) provides the optimal Efficiency-Faithfulness ratio.

| Model | # Params | FLOPs (G) | $\epsilon_{rec} \downarrow$ | $\epsilon_{cf} \downarrow$ | Faithfulness ($F_{\mathbf{x}}$) $\uparrow$ |
|---|---|---|---|---|---|
| TimeSAE (MLP Skeleton) | 11.0M | 0.30 | 0.039 | 0.12 | 1.38 |
| TimeSAE (w/o SE) | 3.3M | 0.42 | 0.021 | 0.07 | 1.98 |
| TimeSAE (1D-CNN) | 3.0M | 0.38 | 0.020 | 0.06 | 2.02 |
| TimeSAE (LSTM+SE) | 5.0M | 0.60 | 0.017 | 0.05 | 2.09 |
| TimeSAE (Transformer+SE) | 8.2M | 1.05 | **0.015** | **0.05** | 2.11 |
| **TimeSAE (TCN+SE)** | **3.5M** | **0.45** | 0.016 | **0.05** | **2.12** |

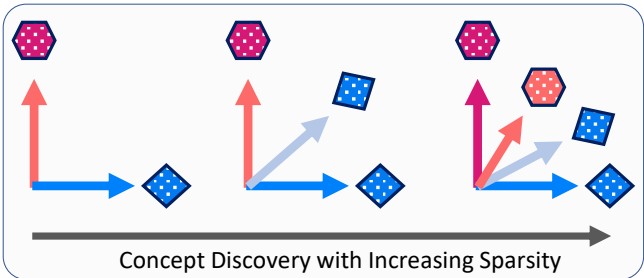

*Figure 12.* Intuition behind the effect of sparsity.

### B.13. Further ablation experiments

#### B.13.1. ABLATION A1 - SAEs SHOULD BE SPARSE, BUT NOT TOO SPARSE.

We investigate the effect of the latent dimension $r$, which determines the number of concept units in TimeSAE and indirectly influences explanation sparsity, as analyzed in our ablation study in the main paper. Sparsity aids in identifying concepts, as illustrated in Figure 12. As shown in Figure 8, increasing $r$ from low values (e.g., 1.3) up to around 1.5-1.6 leads to consistent improvements across all evaluation metrics. This indicates that a moderately larger concept space allows the model to discover richer and more meaningful structures, resulting in better explanations. However, as $r$ continues to increase (e.g., toward 2.0 or more), performance drops, likely due to reduced sparsity and the introduction of redundant or noisy concepts. Although $r$ does not directly encode sparsity, it modulates how selectively the model can activate concept units. A smaller $r$ naturally enforces stronger selection, while a larger $r$ can dilute sparsity. These findings suggest that there is a sweet spot for concept dimensionality, where representations are expressive enough to explain predictions but sparse enough to remain interpretable. We further analyze the sensitivity to the parameter $\alpha$ in Figure 13 for both TimeSAE-JumpReLU and TimeSAE-TopK.

#### B.13.2. ABLATION A2 - TopK PREVENTS ACTIVATION SHRINKAGE

This ablation study investigates how the use of TopK selection in the TimeSAE architecture mitigates the issue of activation shrinkage, which commonly leads to "dead" concepts that cease to learn during training. As shown in Figure 11, employing a fixed, high $\gamma$ value with hard Top-K results in many concept activations collapsing to near zero over training steps, indicating that these concepts become inactive and contribute little to the model's interpretability or performance. In contrast, the soft Top-K variant with a scheduled $\gamma$ that gradually decreases from 15 to 1 maintains more evenly distributed and dynamic concept activations, thereby preventing concept "death." This gradual sparsification approach ensures that concepts remain responsive and learnable throughout training. Complementing this, Figure 9 visually demonstrates that the explanations generated by TimeSAE-TopK produce more focused and interpretable saliency maps compared to other black-box methods such as IG, DynaMask, and TimeX variants. These results collectively highlight that the TopK mechanism with $\gamma$ scheduling not only preserves concept vitality by preventing activation shrinkage but also enhances the clarity and quality of explanations in Transformer-based time series models.

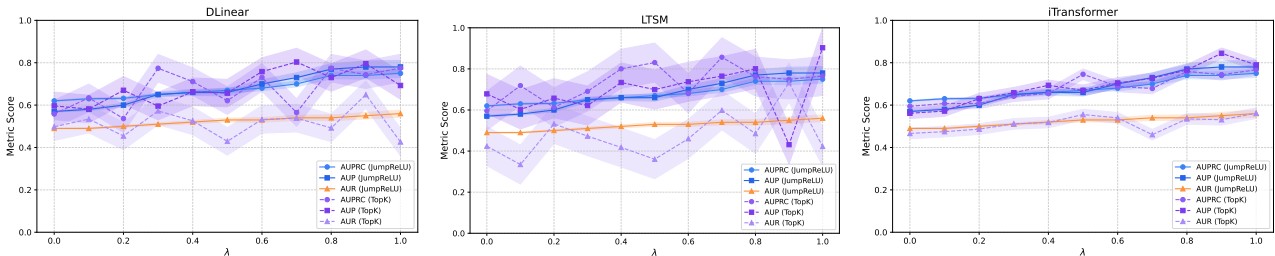

*Figure 13.* Extended ablation study on the effect of the concept consistency weight $\alpha$ for the EliteLJ dataset, evaluating metrics AUPRC, AUP, and AUR across TimeSAE models. Left: DLinear, most metrics perform well at $\alpha = 0.9$. Middle: LSTM, shows similar behavior to DLinear. Right: iTransformer, performance improves as $\alpha$ increases. Solid lines represent TimeSAE-TopK, dashed lines represent JumpReLU, and shaded areas indicate standard deviations over 10 runs. Slightly higher $\alpha$ values lead to more robust explanations.

### B.14. Ablation A3: Out-of-Distribution Robustness

We conduct an ablation study to evaluate the robustness of explanation methods under distribution shift. Specifically, we compare in-distribution (ID) and out-of-distribution (OOD) settings to assess how well different approaches preserve distributional alignment and explanation quality when applied to unseen data. Panel (a) of Table 18 characterizes the degree of distribution shift using kernel density estimation (KDE), Kullback–Leibler divergence (KL), and maximum mean discrepancy (MMD). Panel (b) reports the performance of all methods under these settings, measuring both alignment with the original data distribution and explanation faithfulness. This analysis highlights the ability of each method to generalize beyond the training distribution while maintaining reliable and interpretable explanations.

*Table 18.* Distributional alignment and performance evaluation for in-distribution (ID) and out-of-distribution (OOD) settings. (a) Dataset definition and statistics. (b) Evaluation of method performance. Higher is better for KDE, AUPRC, and $\mathcal{F}_{\mathbf{x}}$; lower is better for KL and MMD.

(a) Setting & Statistics

| Definition & Statistics | | | |
|---|---|---|---|
| **Dataset Pair** | **KDE** | **KL** | **MMD** |
| ● ID: ETTh1 → ETTh1-val trained on ETTh1, and tested on ETTh1-val | $-0.110 \pm 0.01$ | $0.039 \pm 0.002$ | $0.014 \pm 0.001$ |
| ○ OOD: ETTh1 → ETTh2 trained on ETTh1, and tested on ETTh2 | $-0.295 \pm 0.02$ | $0.421 \pm 0.03$ | $0.161 \pm 0.01$ |

(b) Evaluation in OOD settings

| Method | Setting | KDE↑ | KL↓ | MMD↓ | AUPRC↑ | $\mathcal{F}_{\mathbf{x}}$↑ |
|---|---|---|---|---|---|---|
| IG | ● ID | $-46.10_{\pm 1.3}$ | $0.295_{\pm 0.025}$ | $0.027_{\pm 0.004}$ | $0.422_{\pm 0.045}$ | $1.38_{\pm 0.06}$ |
| | ○ OOD | $-49.45_{\pm 1.6}$ | $0.355_{\pm 0.032}$ | $0.120_{\pm 0.012}$ | $0.394_{\pm 0.03}$ | $1.31_{\pm 0.07}$ |
| TimeX | ● ID | $-45.30_{\pm 1.2}$ | $0.288_{\pm 0.02}$ | $0.024_{\pm 0.003}$ | $0.416_{\pm 0.04}$ | $1.35_{\pm 0.05}$ |
| | ○ OOD | $-50.82_{\pm 1.5}$ | $0.342_{\pm 0.03}$ | $0.115_{\pm 0.01}$ | $0.401_{\pm 0.035}$ | $1.28_{\pm 0.06}$ |
| TimeX++ | ● ID | $-44.12_{\pm 1.1}$ | $0.198_{\pm 0.02}$ | $0.019_{\pm 0.002}$ | $0.714_{\pm 0.05}$ | $1.75_{\pm 0.07}$ |
| | ○ OOD | $-48.77_{\pm 1.3}$ | $0.265_{\pm 0.03}$ | $0.101_{\pm 0.01}$ | $0.622_{\pm 0.04}$ | $1.70_{\pm 0.08}$ |
| TimeSAE (Ours) | ● ID | $-43.55_{\pm 1.1}$ | $0.182_{\pm 0.01}$ | $0.016_{\pm 0.002}$ | $0.741_{\pm 0.05}$ | $2.12_{\pm 0.05}$ |
| | ○ OOD | $\mathbf{-47.21_{\pm 1.3}}$ | $\mathbf{0.245_{\pm 0.02}}$ | $\mathbf{0.089_{\pm 0.01}}$ | $\mathbf{0.641_{\pm 0.03}}$ | $\mathbf{2.09_{\pm 0.06}}$ |

### B.15. Ablation A4: Sparsity Efficiency and Activation Frequency

To better understand how different sparsity mechanisms affect explanation quality, we analyze the trade-off between sparsity level and reconstruction fidelity, as well as the resulting activation patterns of learned concepts. In particular, we compare the proposed JumpReLU-based sparsification against the commonly used TopK strategy across varying sparsity budgets. This analysis aims to assess not only reconstruction performance under increasing sparsity constraints, but also the stability and utilization of learned concepts, which are critical for producing interpretable and reliable explanations.

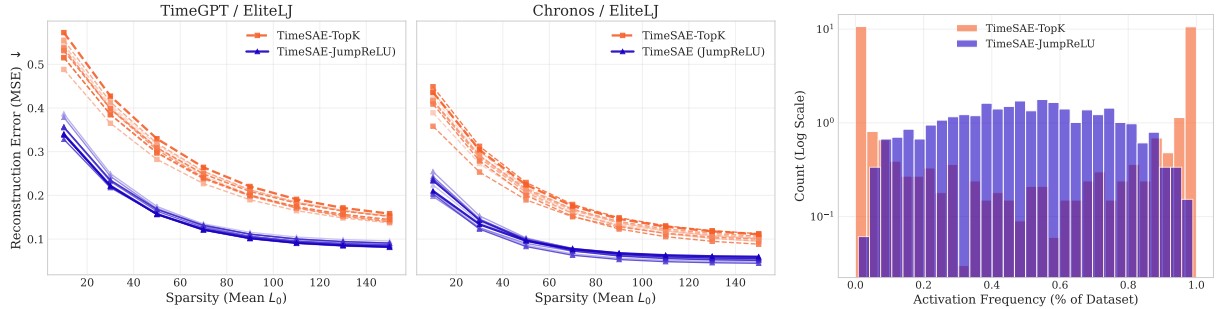

*Figure 14.* Sparsity efficiency and activation frequency. **Left:** TimeGPT on EliteLJ. **Middle:** Chronos on EliteLJ. In both, TimeSAE with JumpReLU outperforms TopK at all $L_0$, showing better sparsity-fidelity trade-offs. **Right:** Log-scale activation histogram: TopK spikes near $0\%$ (*dead concept*), JumpReLU is more distributed; 10 seeds shown in gradient colors.

### B.16. Implementations

#### B.16.1. Pseudo Code of TimeSAE.

```python
def timesae_jumprelu(params, x, sparsity_coefficient, use_pre_enc_bias):
    """
    Computes the forward pass and total loss
    for a JumpReLU-based Sparse Autoencoder.

    Args:
        params: Object containing model parameters
        (weights, biases, log threshold).
        x: Input batch (tensor).
        sparsity_coefficient: Scaling factor
        for the sparsity regularization term.
```

```
12          use_pre_enc_bias: Boolean indicating
13          whether to subtract decoder bias from input.
14
15      Returns:
16          Scalar representing the mean loss over
17          the input batch.
18      """
19      if use_pre_enc_bias:
20          x = x - params.b_dec
21
22      pre_activations = relu(x @ params.W_enc + params.b_enc)
23
24      # Compute threshold from the learnable
25      log value threshold = exp(params.log_threshold)
26
27      # Apply JumpReLU for sparsity-aware feature
28      extraction feature_magnitudes = jumprelu(pre_activations, threshold)
29      # decoding
30      x_reconstructed = feature_magnitudes @ params.W_dec + params.b_dec
31
32      # Compute reconstruction loss
33      reconstruction_error = x - x_reconstructed
34      reconstruction_loss = sum(reconstruction_error ** 2, axis=-1)
35
36      # Compute L0-style sparsity penalty
37      l0_penalty = sum(step(feature_magnitudes, threshold), axis=-1)
38      sparsity_loss = sparsity_coefficient * l0_penalty
39
40      total_loss = mean(reconstruction_loss + sparsity_loss, axis=0)
41      return total_loss
```

### B.16.2. PSEUDO CODE OF TIMESAE (CASE WITH TOPK)

```
1  def gamma_scheduler(step, max_gamma=10.0, min_gamma=1.0, total_steps=10000):
2      """
3      Exponential decay scheduler for $\gamma$.
4      """
5      progress = min(step / total_steps, 1.0)
6      return max_gamma * (min_gamma / max_gamma) ** progress
7
8  def timesae_soft_topk(x, params, k, gamma, sparsity_coeff, use_pre_enc_bias):
9      """
10     Full loss computation for TimeSAE-TopK_gamma.
11     Combines encoding, decoding, and loss into one compact function.
12     Args:
13         x: input batch [batch, features]
14         params: model parameters (W_enc, b_enc, W_dec, b_dec)
15         k: top-k features to retain
16         $\gamma$: current $\gamma$ value (>1 early in training, $\rightarrow$ 1)
17         sparsity_coeff: weight for L0 sparsity loss
18         use_pre_enc_bias: if True, subtract b_dec before encoding
19     Returns:
20         Mean total loss (MSE + sparsity)
21     """
22     # Optional pre-encoder bias
23     if use_pre_enc_bias:
24         x = x - params.b_dec
25
26     # Encode with ReLU
27     pre_activations = relu(x @ params.W_enc + params.b_enc)
28
29     # Compute $\gamma$-scaled TopK mask
30     sorted_vals, _ = torch.sort(pre_activations, dim=-1, descending=True)
31     threshold = sorted_vals[:, k - 1:k]  # shape [batch, 1]
32     topk_mask = (pre_activations >= threshold).float()
33     soft_mask = torch.clamp(gamma * topk_mask, max=1.0)
```

```
34
35      # Apply sparsity
36      sparse_features = pre_activations * soft_mask
37
38      # Decode
39      x_reconstructed = sparse_features @ params.W_dec + params.b_dec
40
41      # Compute losses
42      reconstruction_loss = ((x - x_reconstructed) ** 2).sum(dim=-1)
43      sparsity_loss = sparsity_coeff * (sparse_features > 0).sum(dim=-1)
44      return (reconstruction_loss + sparsity_loss).mean()
45
```

## C. Limitations

While `TimeSAE` provides faithful and interpretable explanations, several practical aspects offer exciting avenues for future research for the time series community:

[1] **Dependence on the Dataset Used to Train the Black-Box Models:** Our method benefits from access to sufficiently large datasets to effectively explain black-box models. In scenarios where data are limited, exploring strategies such as domain adaptation or leveraging similar but different distributions could enable `TimeSAE` to generalize well with fewer data. Developing such approaches can broaden applicability to data-scarce or specialized domains, opening up valuable research directions. We believe that relaxing certain assumptions can be a significant step toward developing more general explainable methods for time series. Moreover, the proposed approach offers a new perspective by leveraging Sparse Autoencoders (SAEs) as an explainability tool for time series, similar to their successful use in large language models for discovering highly interpretable concepts (Huben et al., 2024).

[2] **Sensitivity to Hyperparameter Settings:** Although hyperparameter choices like sparsity levels and dictionary size significantly impact performance, this also presents an opportunity to develop more automated, data-driven tuning methods. Advances in hyperparameter optimization or self-regularizing architectures could reduce manual effort and improve `TimeSAE` 's ease of use and transferability to new tasks.

## D. Reproducibility

All source code for data preprocessing, training, and evaluation is available at https://oublalkhalid.github.io/TimeSAE/. Experiments were conducted using both TPU and GPU hardware, with fixed random seeds to ensure reproducibility across runs. Training was primarily implemented in JAX on TPUs, while equivalent PyTorch implementations were run on NVIDIA A100 GPUs to validate consistency and assess computational trade-offs. The repository includes detailed instructions, preprocessed datasets, and pretrained model checkpoints to enable full reproduction of our results.

