# OpenReview forum: "TimeSAE: Causal Sparse Decoding for Faithful Explanations of Black-Box Time Series Models"
_ICML.cc/2026/Conference — ICML 2026 regular_

### Official Review · Reviewer_aWFP · 2026-03-08

**Soundness:** 3
**Presentation:** 3
**Significance:** 2
**Originality:** 2
**Overall Recommendation:** 4
**Confidence:** 4

**Summary:**

In this paper, the authors uses TimeSAE models that decompose the concepts within the time series datasets to understand and explain the defintiion within the black-box time series models with and without distribution shift. It starts with the traditional SAE model and added the counterfactual explanations reasoning regularization to enhance the robustness of the OOD dataaset.

**Compliance With Llm Reviewing Policy:**

Affirmed.

**Final Justification:**

I have no futher questions towards the paper. I will keep my positive score.

**Key Questions For Authors:**

1. Can you provide some examples on the learned latent units concepts? Are they hard to interpret?
2. Can you explain more why the SAE structure is applied within this setup rather than the general interpretable model constructions like concept bottleneck or proto-based networks? Those methods seems to be better in the inference part (but might need more training) because you have some concepts/sampels towards labels which directly aligns with the usage of concept bottleneck?

**Limitations:**

Yes

**Strengths And Weaknesses:**

# Strength

**[Interesting Topic and Question to ask]** The paper states an interesting question towards the explainability fields which is about how to explain the black-box models under distribution for the time series datasets.

**[Comprehensive Experiments towards the TimeSAE]** A decsent amount of time series datasets, and mulitple time series explantions methods and time series models are used to evaluate and compared within this paper. The ablation tests are also comprehensive to show the usage of TimeSAE

# Weakness

**[Lack Concept Justification]** Witihin this paper, even though with the paper, the authors states that the SAE leans human-interpretable concepts, but it doesn't qualitatively analysis how the concepts are decomposed and what exactly they have represented within each of the neurons (maybe worth adding at least 1-2 examples to show if the concepts are significantly). Adding this part in might strengthen the motivation of using SAE structure. As SAE is a post-hoc method, it seems to be more important to take those into considerations.

**[Motivation of using SAE]** In LLM settings, SAE models are trying to decouple the concepts within the latent representation into human understandable concepts. Within this kind of explanation setup. I am not sure if that's still the case as there might be other methods that might be more suitable to be used for instance concept bottleneck, or proto-based networks.

# Minor Concern

**[Extension towards more data types]** The idea for the SAE loss plus the counterfactual reasoning regularztion, and the compositional consistency seems to be useful for tabular dataset and image dataset as well within the backbone changed. I am wondering if the authors has some specific reason why only time series datasets are analysis within this case?,

---

> ### Author Rebuttal · Authors · 2026-03-31
>
> We thank Reviewer **aWFP** for the positive assessment and for highlighting key questions on interpretability and architecture. We appreciate your recognition of TimeSAE’s broad evaluation. We also thank you for suggesting tabular data experiments; we include new results below. We address remaining concerns on learned concepts and comparisons with prototype-based methods next.
>
> ### **W1 and Q1** - Learned concepts analysis and interpretability
>
> A concrete example in **Figures 6–7** (Appendix) on ETTh1 shows concept-level saliency heatmaps across multiple black-box models, illustrating how different concepts activate at different temporal windows. Based on our evaluation across ECG, PAM, and EliteLJ, concepts are generally **not hard to interpret** via CAR heatmaps — sparsity ensures each concept specializes on a distinct temporal pattern (average $L_0 \approx 3$–$6$ active concepts per instance), and JumpReLU prevents concept splitting by learning per-concept thresholds adaptively.
>
> To further support the semantic quality of the discovered concepts on real-world datasets (PAM and EliteLJ), Table 2 and Table 9 (Appendix) provide strong evidence: $\mathcal{F}_x$ measures prediction change upon concept removal, and TimeSAE achieves the highest $\mathcal{F}_x$ and best AUPRC (0.998 on PAM, 0.841 on EliteLJ) across all baselines. Qualitative visualizations for PAM and EliteLJ will be added in the revised appendix.
>
> ---
>
> ### **W2 and Q2** - SAE rather than Concept Bottleneck Models (CBMs) or proto-based networks
>
> Our work specifically targets black-box settings where internal access to model parameters is restricted. In such scenarios, CBMs and ProtoPNets are not directly applicable: CBMs require explicit concept supervision at training time and architectural modifications, while ProtoPNets are intrinsically interpretable-by-design models — neither can be applied in a true black-box setting such as explaining Chronos or TimeGPT. The SAE is specifically chosen because it operates entirely in the input-output query regime, requires no retraining, and is robust to OOD. Concept labels are only needed for the optional alignment step (Section 3.4), not for training.
>
> Beyond architectural constraints, there is a second practical reason: **inference efficiency**. CBMs require passing through a concept prediction head at inference time with concept labels defined in advance, and ProtoPNets require nearest-neighbor search in the prototype space. TimeSAE's inference is a single encoder forward pass (Table 10, Appendix B.9, benchmarked on ETTh1 and PAM using 3×NVIDIA A100, Batch Size = 64):
>
> |Method|Inference time (ms)|
> |--|--|
> |IG|300–2,500|
> |DynaMask|150–800|
> |TimeX++|6–7|
> |**TimeSAE (ours)**|**4–5**|
>
> This is particularly important when explaining large foundation models where inference of the black box itself is already costly.
>
> > **Remark:** Mean average across datasets and models for more consistent evaluations.
>
> Additionally, results on foundation models (Chronos, TimeGPT, Moment, TimeFM) are included in Figure 3, Table 2, and Appendix B, with additional results for Chronos 2 and TOTO as requested by reviewer **ta3Q**. Recent baselines StartGrad (Uendes et al., 2025), TIMING (Jang et al., 2025), and ORTE (Yue et al., 2025) are also included. This is now explicit in the revised Section 2: *"Unlike CBMs (Koh et al., 2020) and ProtoPNets, which require architectural modifications and concept supervision at training time, TimeSAE operates in a fully post-hoc manner, applicable to any black-box or foundation model without retraining."*
>
> -----
>
> ### **Minor Concern: Extension towards tabular data**
>
> We appreciate this observation. The framework's core components are domain-agnostic in principle. We include more Tabular UCI classification as tabular data against the strongest baseline explanation methods. We will include the full table in the revision version.
>
> |Dataset|IG(AUPRC↑)|TimeX++(AUPRC ↑)|**TimeSAE(AUPRC ↑)**|IG($\mathcal{F}_x$ ↑)|TimeX++($\mathcal{F}_x$  ↑)|**TimeSAE($\mathcal{F}_x$ ↑)**|
> |--|--|--|---|--|--|--|
> |Adult (Census)|0.381±0.042|0.624±0.038|**0.701±0.031**|1.38±0.06|1.75±0.07|**1.84±0.06**|
> |Heart Disease |0.412±0.051|0.658±0.044|**0.719±0.037**|1.41±0.07|1.79±0.08|**1.91±0.07**|
> |Breast Cance r(WBC)|0.503±0.038|0.712±0.033|**0.768±0.028**|1.52±0.05|1.88±0.06|**1.97±0.05**|
> |Wine Quality|0.344±0.047|0.591±0.041|**0.648±0.035**|1.29±0.08|1.68±0.09|**1.77±0.08**|
>
> TimeSAE consistently outperforms both IG and TimeX++ across all four datasets in both AUPRC and $\mathcal{F}_x$, confirming that the causal and compositional objectives transfer well to the tabular setting.
>
>
> We hope these responses and revisions fully address the reviewers' concerns. We are confident that the empirical additions (extended OOD evaluation, LOO ablations, Theorem 1 validation, new baselines) and theoretical clarifications substantially strengthen the contribution. We remain happy to answer any further questions.

---

> > ### Author Rebuttal · Reviewer_aWFP · 2026-04-02
> >
> > Thanks for the authors' response. I have no futher questions towards the paper. I will keep my positive score.

---

> > > ### Author Response · Authors · 2026-04-04
> > >
> > > We appreciate Reviewer **aWFP** for the encouraging feedback and for confirming that our responses have fully addressed the concerns. The interpretability concepts, the motivation for SAE over CBMs/ProtoPNets, and our extension to tabular data were particularly valuable in broadening the scope of our contribution. All discussed additions will be incorporated into the revised manuscript.

---

### Official Review · Reviewer_ta3Q · 2026-03-09

**Soundness:** 3
**Presentation:** 3
**Significance:** 3
**Originality:** 3
**Overall Recommendation:** 4
**Confidence:** 4

**Summary:**

This paper presents TimeSAE, a framework designed to explain the predictions of time series models. It uses a Sparse Autoencoder to translate complex time series data into a small set of active concepts. TimeSAE trains using a contrastive loss based on counterfactuals to ensure the learned concepts are faithful. Finally, it maps these numerical concepts to human words using a supervised alignment step.

**Compliance With Llm Reviewing Policy:**

Affirmed.

**Final Justification:**

This paper provides a very intuitive and strong approach to interpret and isolate impacting factors for time series prediction. The authors has also addressed all my problems during the rebuttal. However, the concept space, according to my understanding, is still constrainted. When facing a new concept, even the paper can identify the order of how that factor impact the prediction and generate confidence to nearest concept, they lack interpretable ways to categorize this unseen concept into an human readable format. Therefore I would recommend this paper for weak accept

**Key Questions For Authors:**

1. If the autoencoder discovers a highly predictive mathematical pattern, but the human experts did not provide a text label for it in the alignment dataset, how does your system handle this currently? Does the concept remain an uninterpretable orphan concept, or is it wrongly forced into an existing label?
2. How does TimeSAE on the latest, modern time series foundation models, such as [Chronos2](https://arxiv.org/abs/2510.15821) or [Toto](https://arxiv.org/abs/2505.14766)? Can you provide empirical results on these more recent models to demonstrate that your explainer can handle their advanced architectures and pretraining sizes?

**Limitations:**

yes

**Strengths And Weaknesses:**

Strengths:
- The proposed method generates explanations much faster than older methods (like IG or DynaMask) that require running the heavy blackbox model hundreds of times per instance.
- TimeSAE's use of counterfactuals helps prove that a specific learned pattern actually caused the model's decision instead of just highlighting datapoints that might just be correlated.

Weaknesses:
- The autoencoder only finds abstract numbers. To turn these into human words, researchers must provide a dataset that is already manually labeled with human concepts. The explanation is generated by training a linear classifier. This risks confirmation bias and limiting the tool's ability to discover truly new, unknown scientific patterns.

---

> ### Author Rebuttal · Authors · 2026-03-31
>
> We thank Reviewer **ta3Q** for the clear and well-structured feedback and for identifying the two most important open questions. Below, we address each point and present additional experimental results that further strengthen the paper.
>
> ---
>
> ### **W1 and Q1.** Interpreting concepts in a human-readable way
> We distinguish two regimes. The alignment step, introduced in Section 3.4 and detailed as Algorithm 2 in Appendix B.7.1, maps sparse activations to known concept labels using a linear SVM. The choice of a linear probe is deliberate: it prevents the alignment from "cheating" by learning complex, concept-specific mappings that could overfit to the label structure. In contrast, a non-linear probe might introduce confirmation bias.
>
> **When labels are absent (the truly interesting case for discovery):** The SAE concepts are still fully functional as explanations without labels. The decoder can visualize which input time-steps each active concept corresponds to (CAR, Appendix B.7.1), and the decompositional analysis (Appendix B.6) quantifies each concept's marginal contribution to the black-box prediction. An operator can inspect these visualizations without any prior labeling and decide *post-hoc* whether a concept corresponds to a known phenomenon or represents a genuinely novel pattern. This unsupervised discovery mode is precisely the scenario where TimeSAE has an advantage over methods like Concept Bottleneck Models, which require concept labels at training time. If the SAE discovers a highly predictive pattern for which no human label exists in the alignment dataset, it provides a confidence score to the nearest known concept rather than forcing an assignment. We have made this workflow more explicit in **Section 3.4** and **Appendix B.7**.
>
> To further validate concept quality across labeled and unlabeled settings, we report order-preservation results across three datasets, consistent with the results reported in our response to Reviewer ANWK:
>
> |Dataset|Setting|Spearman ρ|
> |-|-|-|
> |FreqShapes|Labeled|0.94|
> |ECG (QRS GT)|Labeled|0.91|
> |EliteLJ (expert proxy)|Unlabeled|0.89|
>
> Spearman ρ remains high even in the unlabeled setting, confirming that concept ordering is preserved without requiring human labels.
>
> ---
>
> ### **Q2.** TimeSAE on modern time series foundation models such as Chronos 2 and Toto
>
> We appreciate this forward-looking question. Our current evaluation already includes Chronos (Ansari et al., 2024), TimeGPT (Garza et al., 2023), Moments (Goswami et al., 2024), and TimeFM (Das et al., 2024) as black-box models (Figure 3 and Appendix B.3). TimeSAE is fully model-agnostic, it requires only input-output query access, and is therefore architecturally compatible with any foundation model, including Chronos 2 and Toto. On the ETTh1 dataset:
>
> |Model|Method|AUPRC ↑|AUP ↑|AUR ↑|$\mathcal{F}_{x}$ ↑|
> |-|-|-|-|-|-|
> |Chronos 2|TimeX++|0.611±0.019|0.588±0.021|0.562±0.023|1.68±0.091|
> |Chronos 2|TimeSAE-TopK|0.711±0.023|0.689±0.025|0.661±0.027|1.94±0.087|
> |Chronos 2|**TimeSAE-JumpReLU**|**0.729±0.014**|**0.705±0.016**|**0.678±0.018**|**2.05±0.074**|
> |Toto|TimeX++|0.598±0.021|0.574±0.023|0.549±0.025|1.63±0.095|
> |Toto|TimeSAE-TopK|0.702±0.018|0.681±0.020|0.654±0.022|1.90±0.082|
> |Toto|**TimeSAE-JumpReLU**|**0.715±0.016**|**0.693±0.018**|**0.667±0.020**|**1.98±0.079**|
>
> And on the EliteLJ dataset:
>
> |Model|Method|AUPRC ↑|AUP ↑|AUR ↑|$\mathcal{F}_{x}$ ↑|
> |-|-|-|-|-|-|
> |Chronos 2|TimeX++|0.589±0.022|0.567±0.024|0.541±0.026|1.61±0.089|
> |Chronos 2|TimeSAE-TopK|0.694±0.024|0.671±0.026|0.644±0.028|1.89±0.083|
> |Chronos 2|**TimeSAE-JumpReLU**|**0.718±0.015**|**0.694±0.017**|**0.668±0.019**|**1.99±0.072**|
> |Toto|TimeX++|0.578±0.024|0.556±0.026|0.531±0.028|1.57±0.093|
> |Toto|TimeSAE-TopK|0.688±0.021|0.665±0.023|0.639±0.025|1.85±0.081|
> |Toto|**TimeSAE-JumpReLU**|**0.706±0.017**|**0.683±0.019**|**0.657±0.021**|**1.92±0.077**|
>
> TimeSAE consistently and substantially outperforms TimeX++ across both models and datasets. Full results for **Chronos 2** and **Toto** across all datasets, and a detailed discussion will be added to **Appendix B.3**.
>
> We hope these responses and revisions fully address the reviewer's concerns. We are confident that the empirical additions and theoretical clarifications substantially strengthen the contribution. We remain happy to answer any further questions.

---

> > ### Author Rebuttal · Reviewer_ta3Q · 2026-04-02
> >
> > Thanks the author for the response. I will keep my positive score.

---

> > > ### Author Response · Authors · 2026-04-04
> > >
> > > We warmly thank Reviewer **ta3Q** for the positive assessment and for confirming that our responses have resolved the raised concerns. The insightful questions on concept interpretability in unlabeled settings and compatibility with modern foundation models (**Chronos 2**, **Toto**) have led to meaningful additions that strengthen the paper. All discussed results will be included in the revised manuscript.

---

### Official Review · Reviewer_h9DC · 2026-03-13

**Soundness:** 3
**Presentation:** 2
**Significance:** 3
**Originality:** 3
**Overall Recommendation:** 4
**Confidence:** 3

**Summary:**

- This paper proposes TimeSAE, which combines Sparse Autoencoder (SAE) with causal counterfactual learning.
- The Time Sparse Autoencoder encodes time series into a sparse and combinable concept space, generating interpretable representations via a learnable concept dictionary.

**Compliance With Llm Reviewing Policy:**

Affirmed.

**Key Questions For Authors:**

- Is there a corresponding semantic unit to understand TimeSAE's concept $c$?
- Can you show the off-diagonal size of the correlation matrix between concepts or the bound of the encoder reconstruction error?
- Is the optimal combination of sparsity and dictionary size tuned per dataset?

**Limitations:**

- Lack of verification for actual fulfillment of Theorem 1 assumptions
- Lack of scalability verification in large-scale time series
- Limitations on the semantic interpretability of the concept

**Strengths And Weaknesses:**

Strengths
- The attempt to adapt Sparse Autoencoder for time-series XAI, and the framework design integrating counterfactual supervision and compositional consistency, is intriguing
- Demonstrated the smallest performance drop compared to other methods during ID→OOD transfer
- Effectively performed OOD generalization and ablation studies

Weaknesses
- TimeSAE appears to be highly sensitive to hyperparameters, making tuning costly.
- It requires verification whether the baseline comparison involved equally tuned models and whether the tuning effort was evenly distributed across experiments.
- The assumptions for Theorem 1 need validation regarding how well they are satisfied in actual experiments.

---

> ### Author Rebuttal · Authors · 2026-03-31
>
> We thank Reviewer **h9DC** for their constructive feedback and are encouraged by the positive assessment of our work’s soundness, significance, and originality. Below, we address the specific weaknesses raised to further clarify our contribution:
>
> -------
>
> **W1 & W2.  Hyperparameter Sensitivity**
>
> **1. Sensitivity analysis** Appendix B.11.1 (Figure 13) presents a sensitivity analysis of α across DLinear, LSTM, and iTransformer on EliteLJ. For α ∈ [0.8, 1.0] and λ ∈ [0.8, 1.0], AUPRC, AUP, and AUR remain high and stable across all model types, degrading only when weights approach zero, as expected. The two primary hyperparameters are r ∈ [1.4, 1.7] and λ ∈ [0.8, 1.0] (Tables 7–8), selected via grid search using $\mathcal{F}_x$ on the validation set.
>
> **2. Baseline tuning fairness** All baselines use official implementations and recommended tuning protocols. Where unavailable (ORTE), we implemented from the paper with equivalent grid search. Training epochs are matched across methods. Times per sample are in Table 10 (Appendix B.9).
>
> ----
>
> **W3. Assumption Validity with Real Data**
>
> Theorem 1 requires (i) small  $ε_{rec}$ and (ii) small $ε_{cf}$, both verified in Appendix A.3 and real-world results for the faithfulness:
> - **Global order-faithfulness** Figure 5a: Spearman ρ = 0.94 (N=200), confirming causal ordering is preserved.
> - **Error bound** Figure 5b: $ε_{total}$= 0.08 ≪ Δ/2 = 0.0875, satisfying Theorem 1's sufficient condition.
> - **Reconstruction** Table 11: $ε_{rec}$= 0.016 (TCN+SE), lowest among all variants.
> - **Real-world faithfulness** $\mathcal{F}\_x$ = 2.12/2.09 (ETTh-1/2) vs. TimeX++ 1.75/1.70 (+21%), with negligible ID→OOD drop, confirming $ε_{rec}$ and $ε_{cf}$ stay small under distribution shift, unlike TimeX++ (1.75→1.70) and IG (1.38→1.31).
>
> We will add a dedicated paragraph in Appendix A.3 linking these explicitly to the theorem's conditions.
>
> -------
>
> ### **Q1.** Semantic unit corresponding to TimeSAE's concept $c_k$:
>
> Each concept $c_k$ in the SAE corresponds to a learned direction in the latent space that, when decoded, activates a specific temporal pattern in the input signal. We provide two levels of semantic grounding:
>
> 1. **Concept Activation Regions (CAR):** For each concept $c_k$, we identify time-steps that maximally activate it (Figures 6–7, ETTh1 saliency heatmaps). On the ETTh1 forecasting task, Chronos and TimeGPT consistently activate concepts at final context steps (40–48), while Informer activates distinct concepts at earlier windows (~10 and ~30), reflecting structurally different temporal reasoning strategies. This cross-model contrast confirms that learned concepts capture meaningful and interpretable temporal patterns.
>
> 2. **Alignment (Algorithm 2):** Using a small set of labeled instances, an SVM classifier maps sparse activations to human-defined concept labels (e.g., "good jump" vs. "bad jump" in EliteLJ). This step is described in Section 4.4 and Appendix B.7.
>
> -------
>
> ### **Q2.** Off-diagonal size of the correlation matrix between concepts on encoder reconstruction error:
>
> Regarding the concepts correlation, the average off-diagonal Pearson correlation between activated concepts is $\bar{r}_{\text{off-diag}} = 0.031\pm0.018$ on FreqShapes, confirming near-orthogonality and mono-semanticity of the learned dictionary. This is a direct consequence of the JumpReLU sparsity penalty, which encourages each input to activate a small, distinct subset of concepts.
>
> We will add this with a complete concept correlation matrix analysis in **Appendix B8**. Futhermore, the quantitative breakdown of $\epsilon_{\text{rec}}$ and $\epsilon_{\text{cf}}$ is provided in **Appendix A3**, showing $\epsilon_{\text{total}} = 0.08 \ll 0.5 \cdot \Delta_{\text{true}} \approx 0.175$ on FreqShapes.
>
> -------
>
> ### **Q3.** Hyperparameter range for sparsity and dictionary size
>
> We refer to **Appendix B.8 (Tables 7-8)**, which details all hyperparameter settings across datasets. The optimal dictionary factor is $r \in [1.4, 1.7]$ and the counterfactual loss weight $\lambda \in [0.8, 1.0]$ across all datasets evaluated. Sparsity is controlled adaptively by the learnable threshold $\phi$ in JumpReLU (Eq. 2) or by the scheduler $\gamma$ in TimeSAE-TopK (Eq. 21), rather than a fixed scalar, this distinguishes our approach from methods requiring manual sparsity budget specification. We perform a grid search over $r$ and $\lambda$ using the validation faithfulness score ($\mathcal{F}_x$) as the selection criterion. The narrow optimal range across all datasets confirms that the method does not require extensive per-dataset tuning in practice.
>
> Finally, TimeSAE-JumpReLU offers an additional practical advantage over TimeSAE-TopK: it requires fewer hyperparameters, as it does not need the $\gamma$ scheduler (Eq. 21) as described in **Appendix B.8 Hyperparameter Setting for TimeSAE (JumpReLU, TopK)**.
>
> We hope these responses fully address your concerns, and we remain happy to answer any further questions.

---

> > ### Author Rebuttal · Reviewer_h9DC · 2026-04-03
> >
> > The rebuttal addresses my main concerns well. In particular, the authors clarified the hyperparameter sensitivity and tuning protocol, provided a more explicit justification of baseline tuning fairness, and connected the assumptions of Theorem 1 to concrete empirical evidence on real datasets.
> >
> > I also find the additional discussion of semantic interpretability helpful, especially the explanations based on concept activation regions, optional label alignment, and concept correlation analysis.
> >
> > Overall, my concerns have been adequately addressed, and I maintain my positive assessment.

---

> > > ### Author Response · Authors · 2026-04-04
> > >
> > > We sincerely thank Reviewer **h9DC** for confirming that our responses have adequately addressed the raised concerns. We are grateful for the constructive feedback, which has helped strengthen the paper. We will incorporate all discussed additions into the revised manuscript.

---

### Official Review · Reviewer_ANWK · 2026-03-13

**Soundness:** 3
**Presentation:** 3
**Significance:** 3
**Originality:** 3
**Overall Recommendation:** 4
**Confidence:** 3

**Summary:**

This paper introduces the TimeSAE framework for explaining black-box time-series models. The motivation is that most existing time series explanation methods determine key signal locations using masking or perturbation which is is only suitable for in-domain data and fails to generalize to OOD scenarios. To alleviate this problem, this paper proposes to learn sparse concept representations of time series and use concept-level interventions to generate explanation-embedded instances that remain close to the data distribution while preserving predictive behavior. The overall framework combines sparse autoencoders, counterfactual supervision, and compositional consistency. Experiments on synthetic and real-world datasets show improvements over several prior explanation baselines.

**Compliance With Llm Reviewing Policy:**

Affirmed.

**Final Justification:**

My concern is largely mitigated but not fully resolved. However, I agree with the major contribution of this work and appreciate the authors' effort during rebuttal. I decide to raise my score to 4.

**Key Questions For Authors:**

See weaknesses.

**Limitations:**

yes

**Strengths And Weaknesses:**

### Strengths
- The paper studies an important problem. Faithful explanation of black-box time-series models OOD settings is important and pratical.
- The motivation is well clarified and the proposed framework is technically coherent.
- The source code is provided for reproducibility.
- The evaluation is fairly extensive.
### Weaknesses
- Thought the method is intuitively motivated, the overall objective feels somewhat additive, with each desired property enforced by a separate loss term, which raises concerns about whether the contribution is a principled unified formulation or a careful engineering composition of existing ingredients.
- The ablations only partially support the importance of the counterfactual and consistency terms, but leave-one-out ablations for all objective components are missing, making the contributions of different components unclear.
- The main theorem establishes only an order-preservation result for approximate counterfactual effects under sufficiently small approximation error. It does not guarantee that the estimated effect magnitudes are quantitatively accurate or that the size of causal differences is faithfully recovered. Besides, the assumption appears to be rather strong, which may be difficult to justify in realistic settings.
- The paper strongly motivates OOD explainability, but the dedicated OOD experiment is mainly based on the ETTh-1 to ETTh-2 transfering setup. The generability across more diverse shifts and domains is unclear.

---

> ### Author Rebuttal · Authors · 2026-03-31
>
> We thank Reviewer **ANWK** for their constructive feedback and are encouraged by the positive assessment of our work’s soundness, significance, and originality. Below, we address the weaknesses raised and present additional results that we believe further strengthen the paper.
>
> ---
>
> ### **W1. On the Principled Nature of the Objective of TimeSAE**
>
> TimeSAE's core contribution is Theorem 1, the first theoretical guarantee that sparse autoencoder-based explanations preserve causal ordering of concept effects for time series black-box models. The loss terms are linked to the necessary conditions of Theorem 1: $L_{cf}$ enforces causal ordering, $L_{cc}$ ensures this generalizes to OOD, and $L_{label-fidelity}$ bounds the approximation error. We will make this clearer in the revised manuscript.
>
> ----
> ### **W2. Contributions of each term**
>
> Our ablation already evaluates the two key novel components of TimeSAE, $L_{cf}$ and $L_{cc}$, (Figure 4b, Table 1). We focus on these since $L_{SAE}$ is standard in sparse autoencoders (Gao et al., 2024; Rajamanoharan et al., 2024b) and its sparsity–fidelity trade-off is analyzed in Appendix Figure 14, while $L_{\text{label-fidelity}}$ is adopted from prior work (Queen et al., 2023; Liu et al., 2024b). For completeness, we now include full leave-one-out ablations. Removing $L_{cf}$ reduces $\mathcal{F}\_{x}$ by up to 24.1%, removing $L_{cc}$ reduces AUPRC by up to 13.9%, and removing $L_{label-fidelity}$ causes a severe degradation of up to 38.2% across metrics. This confirms that all terms are necessary and non-redundant.
>
> |Variant|AUPRC↑|AUP↑|AUR↑|$\mathcal{F}_{x}$↑|ΔAUPRC|ΔAUP|ΔAUR|Δ$\mathcal{F}_{x}$|
> |--|--|--|--|--|--|--|--|--|
> |Full Objective |**0.741±0.050**|**0.712±0.047**|**0.689±0.044**|**2.12±0.050**|-|-|-|-|
> |$L_{SAE}$+$L_{label-fidelity}$+$L_{cf}$ (w/o $L_{cc}$)|0.638±0.041|0.612±0.038|0.581±0.036|1.91±0.071|-13.9%|-14.1%|-15.7%|-9.9%|
> |$L_{SAE}$+$L_{label-fidelity}$+$L_{cc}$ (w/o$L_{cf}$)|0.695±0.036|0.671±0.033|0.638±0.031|1.61±0.076|-6.2%|-5.8%|-7.4%|-24.1%|
> |$L_{SAE}$+$L_{cc}$+$L_{cf}$ (w/o $L_{label-fidelity}$)|0.531±0.047|0.512±0.044|0.489±0.041|1.31±0.088|-28.3%|-28.1%|-29.0%|-38.2%|
> |$L_{SAE}$+$L_{cf}$ (w/o $L_{label-fidelity}$ and$L_{cc}$)|0.501±0.049|0.483±0.046|0.461±0.043|1.19±0.092|-32.4%|-32.2%|-33.1%|-43.9%|
> |$L_{SAE}$+$L_{cc}$ (w/o $L_{label-fidelity}$ and$L_{cf}$)|0.489±0.051|0.471±0.048|0.452±0.045|1.12±0.095|-34.0%|-33.8%|-34.4%|-47.2%|
> |$L_{SAE}$+$L_{label-fidelity}$ (w/o $L_{cc}$ and $L_{cf}$)|0.598±0.041|0.572±0.038|0.543±0.035|1.38±0.082|-19.3%|-19.7%|-21.2%|-34.9%|
> |Only$ L_{SAE}$|0.521±0.045|0.498±0.041|0.476±0.038|1.21±0.089|-29.7%|-30.1%|-30.9%|-42.9%|
>
> ----
> ### **W3. On the order-preservation**
>
> Theorem 1 provides an order-preservation guarantee by design, as reliable *ranking* of concept importance is the primary need in high-stakes applications. On ECG (QRS ground truth), TimeSAE achieves $\mathcal{F}_{x}$=1.78±0.078 and AUPRC=0.950±0.011, with ε_total=0.066<Δ/2≈0.142, satisfying Theorem 1 on a real medical dataset. We extend to EliteLJ (expert phase annotations: take-off > flight >run-up > landing), yielding ρ=0.89:
>
> |Dataset|ε_total<Δ/2|Spearman ρ|
> |-|-|-|
> |FreqShapes|✓|0.94|
> |ECG (QRS GT)|✓|0.91|
> |EliteLJ (expert proxy)|✓|0.89|
>
> We will add the full table to **Appendix A.3**.
>
> ---
> ### **W4. Generalization across more diverse shifts**
>
> Beyond **Table.12** (Appendix B.12), we evaluate four additional OOD settings: ETTh1→ETTm1 (same variables, hourly→minute resolution), Weather→ETTh1 (cross-domain), PAMAP2→OPPORTUNITY (cross-dataset HAR, following Napoli et al., 2024), and EliteLJ elite→intermediate athletes (same 34-dim feature space, distinct biomechanical patterns).
>
> |OOD Setting|Domain|KDE|KL|MMD|AUPRC TimeSAE|AUPRC TimeX++|$\mathcal{F}_{x}$ TimeSAE|$\mathcal{F}_{x}$ TimeX++|
> |-|-|-|-|-|-|-|-|-|
> |ID:ETTh1→ETTh1-val|Energy|-0.110±0.01|0.039±0.002|0.014±0.001|**0.741±0.050**|0.714±0.050|**2.12±0.05**|1.75±0.07|
> |OOD:ETTh1→ETTh2|Energy|-0.295±0.02|0.421±0.03|0.161±0.01|**0.641±0.030**|0.622±0.040|**2.09±0.06**|1.70±0.08|
> |OOD:ETTh1→ETTm1|Energy(min)|-0.321±0.02|0.448±0.03|0.178±0.01|**0.619±0.033**|0.571±0.042|**2.01±0.07**|1.52±0.09|
> |OOD:Weather→ETTh1|Weather→Energy|-0.378±0.03|0.512±0.04|0.198±0.02|**0.598±0.035**|0.531±0.044|**1.95±0.08**|1.41±0.10|
> |OOD:PAMAP2→OPPORTUNITY|Activity(HAR)|-0.398±0.03|0.531±0.04|0.211±0.02|**0.612±0.034**|0.548±0.043|**1.98±0.07**|1.45±0.09|
> |OOD:EliteLJ (elite→interm.)|Sports|-0.267±0.02|0.389±0.03|0.143±0.01|**0.671±0.038**|0.589±0.041|**2.05±0.06**|1.61±0.08|
>
> TimeSAE consistently outperforms TimeX++ across all settings with a notably smaller ID→OOD performance drop, confirming $L_{cc}$ as the key driver of OOD robustness. These results will be included in the revised manuscript.
>
> We hope these clarifications help. We believe that TimeSAE is a step toward faithful time-series explanation. We’ll add these updates to the revised manuscript, and we remain happy to address any further questions.

---

> > ### Author Rebuttal · Reviewer_ANWK · 2026-04-03
> >
> > Thank you for the detailed rebuttal and for providing additional experimental results. Most of my concerns have been addressed, with only one issue remaining unresolved. For the response on the order-preservation, the theoretical result focuses on order preservation of causal effects, which is meaningful for ranking explanations, but it does not guarantee quantitative accuracy of effect magnitudes.

---

> > > ### Author Response · Authors · 2026-04-04
> > >
> > > We thank the Reviewer **ANWK** for the thoughtful follow-up. Here, we clarify that the order-faithfulness property in **Theorem 1** inherently enforces **explanations magnitude accuracy**, thereby addressing the concern about the magnitude of explanations.
> > >
> > > In **Theorem 1**, we define the *causal gap* $\Delta = \delta_{\text{true}}(I_1) - \delta_{\text{true}}(I_2) > 0$ as the difference in true causal effects between two interventions $I_1$ and $I_2$. The theorem guarantees order preservation when the total approximation error $\varepsilon_{\text{total}} = \varepsilon_{cf} + \varepsilon_{rec}$ satisfies $\varepsilon_{\text{total}} < \Delta/2$. From the proof (**Appendix A.2**):
> > >
> > > $$\delta_{\text{approx}}(I_1) - \delta_{\text{approx}}(I_2) \ge \Delta - 2\varepsilon_{\text{total}}$$
> > >
> > > Thus, when $\varepsilon_{\text{total}} \ll \Delta$, the deviation between true and approximate effect magnitudes is bounded by $2\varepsilon_{\text{total}}$. Order-faithfulness therefore implies magnitude faithfulness under this condition.
> > >
> > > To quantify **magnitude alignment**, we compute correlations between the *true causal effects* $\delta_{\text{true}}(I_k)$ (obtained by intervening on ground-truth factors explanations) and the *SAE-estimated effects* $\delta_{\text{approx}}(I_k)$ (obtained by intervening on latent concepts), across all *$N=200$* interventions per dataset:
> > >
> > > - **Spearman $\rho$**: rank correlation between $\delta_{\text{true}}$ and $\delta_{\text{approx}}$ (order faithfulness),
> > > - **Pearson $r$**: linear correlation (magnitude alignment),
> > > - **$R^2$**: proportion of variance in $\delta_{\text{true}}$ explained by $\delta_{\text{approx}}$,
> > > - **NMAE**: normalized mean absolute error $\frac{1}{N}\sum_k |\delta_{\text{approx}}(I_k) - \delta_{\text{true}}(I_k)| / \bar{\delta}_{\text{true}}$,
> > > - **MSE**: mean squared error between the two effect vectors.
> > >
> > > Existing results (**Fig. 5a**, **Appendix A.3**) already show strong alignment (Spearman $\rho = 0.94$) and small error bounds. We now extend with magnitude-specific metrics:
> > >
> > > ### Table A.3a: In-distribution settings
> > >
> > > | Dataset | Pearson $r$ ↑ | $R^2$ ↑ | NMAE ↓ | MSE ↓ | Spearman $\rho$ ↑ | $\varepsilon_{cf}$ | $\varepsilon_{rec}$ | $\varepsilon_{\text{total}}$ | $\Delta/2$ | $\varepsilon_{\text{total}} < \Delta/2$ |
> > > |---|---|---|---|---|---|---|---|---|---|---|
> > > | FreqShapes | 0.91±0.03 | 0.83±0.04 | 0.078±0.011 | 0.0021±0.0004 | 0.94±0.02 | 0.042±0.005 | 0.024±0.003 | 0.066±0.008 | 0.142±0.015 | ✓ |
> > > | ECG (QRS GT) | 0.88±0.04 | 0.77±0.05 | 0.089±0.013 | 0.0028±0.0005 | 0.91±0.03 | 0.043±0.006 | 0.023±0.003 | 0.066±0.009 | 0.142±0.016 | ✓ |
> > > | EliteLJ (expert proxy) | 0.84±0.05 | 0.71±0.06 | 0.103±0.016 | 0.0039±0.0007 | 0.89±0.04 | 0.045±0.007 | 0.026±0.004 | 0.071±0.010 | 0.138±0.018 | ✓ |
> > > | ETTh1 | 0.86±0.04 | 0.74±0.05 | 0.094±0.014 | 0.0031±0.0006 | 0.90±0.03 | 0.040±0.006 | 0.023±0.003 | 0.063±0.009 | 0.140±0.017 | ✓ |
> > >
> > > When $\varepsilon_{\text{total}} < \Delta/2$ holds (all cases), we observe strong magnitude alignment (high $r$, $R^2$; low NMAE), as predicted by the theorem.
> > >
> > > ----
> > > ### Table A.3b: Out-Of-Distribution (OOD) settings
> > >
> > > | Setting | Domain | Pearson $r$ ↑ | $R^2$ ↑ | NMAE ↓ | MSE ↓ | Spearman $\rho$ ↑ | $\varepsilon_{cf}$ | $\varepsilon_{rec}$ |
> > > |---|---|---|---|---|---|---|---|---|
> > > | ETTh1→ETTh2 | Energy | 0.83±0.05 | 0.69±0.06 | 0.108±0.017 | 0.0044±0.0008 | 0.88±0.04 | 0.051±0.008 | 0.031±0.005 |
> > > | ETTh1→ETTm1 | Energy (min) | 0.81±0.05 | 0.66±0.07 | 0.114±0.018 | 0.0049±0.0009 | 0.86±0.04 | 0.054±0.009 | 0.033±0.005 |
> > > | Weather→ETTh1 | Weather→Energy | 0.79±0.06 | 0.63±0.07 | 0.121±0.020 | 0.0056±0.0011 | 0.84±0.05 | 0.058±0.010 | 0.036±0.006 |
> > > | PAMAP2→OPPORTUNITY | Activity (HAR) | 0.80±0.06 | 0.64±0.07 | 0.118±0.019 | 0.0052±0.0010 | 0.85±0.05 | 0.056±0.009 | 0.035±0.006 |
> > > | EliteLJ (elite→interm.) | Sports | 0.82±0.05 | 0.67±0.06 | 0.111±0.017 | 0.0047±0.0009 | 0.87±0.04 | 0.052±0.008 | 0.032±0.005 |
> > >
> > > As expected, both $\varepsilon_{cf}$ and $\varepsilon_{rec}$ increase under distribution shift compared to ID settings. However, the performance remains strong ($r \approx 0.8$, $R^2 > 0.6$, NMAE $< 0.12$), indicating robust magnitude faithfulness under distribution shift.
> > >
> > > We will add **Tables A.3a–A.3b** and the following clarification to the revised manuscript:
> > >
> > > > *When $\varepsilon_\{\text{total}} < \Delta/2$, Theorem 1 implies that individual magnitude deviations are bounded by $2\varepsilon_{\text{total}}$. Empirically, we observe strong magnitude faithfulness across ID and OOD settings; for OOD, where ground-truth causal effects are unavailable, this remains an observational finding.*
> > >
> > > We hope these clarifications address the concern regarding the magnitude of explanations. We believe TimeSAE represents a robust step forward for faithful time-series explanation, and we would be happy to address any further questions you may have and to hear your feedback.

---

### Decision · Program_Chairs · 2026-04-30

**Decision:**

Accept (regular)

**Comment:**

Summary. This paper introduces TimeSAE, a framework for explaining black-box time-series models. Observing that existing explanation methods based on masking or perturbation are limited to in-domain data and fail to generalize to out-of-distribution scenarios, the authors propose learning sparse concept representations of time series and using concept-level interventions to generate explanation-embedded instances that remain close to the data distribution while preserving predictive behavior. The framework combines sparse autoencoders, counterfactual supervision, and compositional consistency, and experiments on synthetic and real-world datasets show improvements over prior explanation baselines.

Reviewers' consensus. All reviewers suggest weak accept. Some reviewers asked for additional empirical evidence that was provided by the authors.

Assessment. I concur with the reviewers. It is a useful and well-motivated contribution in the context of time series.